# Speed Up Federated Learning in Heterogeneous Environment: A Dynamic Tiering Approach

## Abstract

Federated learning (FL) enables collaboratively training a model while keeping the training data decentralized and private. However, one significant impediment to training a model using FL, especially large models, is the resource constraints of devices with heterogeneous computation and communication capacities as well as varying task sizes. Such heterogeneity would render significant variations in the training time of clients, resulting in a longer overall training time as well as a waste of resources in faster clients. To tackle these heterogeneity issues, we propose the Dynamic Tiering-based Federated Learning (DTFL) system where slower clients dynamically offload part of the model to the server to alleviate resource constrains and speed up training. By leveraging the concept of Split Learning, DTFL offloads different portions of the global model to clients in different tiers and enables each client to update the models in parallel via local-loss-based training. This helps reduce the computation and communication demand on resource-constrained devices and thus mitigates the straggler problem. DTFL introduces a dynamic tier scheduler that uses tier profiling to estimate the expected training time of each client, based on their historical training time, communication speed, and dataset size. The dynamic tier scheduler assigns clients to suitable tiers to minimize the overall training time in each round. We first theoretically prove the convergence properties of DTFL. We then train large models (ResNet-56 and ResNet-110) on popular image datasets (CIFAR-10, CIFAR-100, CINIC-10, and HAM10000) under both IID and non-IID systems. Extensive experimental results show that compared with state-of-the-art FL methods, DTFL can significantly reduce the training time while maintaining model accuracy.

## 1 Introduction

Federated learning (FL), which allows clients to train a global model collaboratively without sharing their sensitive data with others, has become a popular privacy-preserving distributed learning paradigm. In FL, clients update the global model using their locally trained weights to avoid sharing raw data with the server or other clients. This training process, however, becomes a significant hurdle for training large models when clients are resource-constrained devices (e.g., mobile/IoT devices, and edge servers) with heterogeneous computation and communication capacities in addition to different dataset sizes. Such heterogeneity would incur a significant impact on training time and model accuracy in conventional FL systems (i.e., larger training time is required to reach similar accuracy compared to non-heterogeneous systems) Yang et al. (2021); Abdelmoniem et al. (2023).

To train large models with resource-constrained devices, various methods have been proposed in the literature. One solution is to split the global model into a client-side model (i.e., the first a few layers of the global model) and a server-side model, where the clients only need to train the small client-side model via Split Learning (SL) Gupta & Raskar (2018); Vepakomma et al. (2018). Liao et al. (2023) improves model training speed in split federated learning (SFL) by giving local clients control over both the local updating frequency and batch size. However, in SFL, each client needs to wait for the back-propagated gradients from the server to update its model, and the communication overhead for transmitting the forward/backward signals between the server and clients can be substantial at each training round (i.e., time needed to complete a round of training). To address these issues, He et al. (2020a); Cho et al. (2023) uses a knowledge transfer training algorithm, to train small models at clients and periodically transfer their knowledge via knowledge distillation to a large server-side model. Han et al. (2021) develops a federated SL algorithm that addresses the latency

and communication issues by integrating local-loss-based training into SL. However, the client-side models in He et al. (2020a); Han et al. (2021) are fixed throughout the training process, and choosing suitable client-side models in heterogeneous environments is challenging as the resources of clients may change over time. Another solution is to divide clients into tiers based on their training speed and select clients from the same tier in each training round to mitigate the straggler problem Chai et al. (2020; 2021). However, existing tier-based works Chai et al. (2020; 2021) still require clients to train the entire global model, which is not suitable for training large models.

In this paper, we propose the Dynamic Tiering-based Federated Learning (DTFL) system, to speed up FL for training large models in heterogeneous environments. DTFL aims to not only incorporate benefits from both SFL Han et al. (2021) and tier-based FL Chai et al. (2020), but also address the latency issues and reduce the training time of these works in heterogeneous environments. In DTFL, we divide clients into different tiers. In different tiers, DTFL offloads different portions of the global model from each client to the server. Then each client and the server update the models in parallel using local-loss-based training Nøkland & Eidnes (2019); Belilovsky et al. (2020); Huo et al. (2018); Han et al. (2021). In a heterogeneous environment, the training time of each client can change over time. Static tier assignments can result in severe straggler issues when clients with limited computation and communication resources (e.g., due to other concurrently running applications on mobile devices) are allocated to tiers demanding high levels of resources. To address this challenge, we propose a dynamic tier scheduler that assigns clients to suitable tiers based on their capacities, their task size, and their current training speed. The tier scheduler employs tier profiling to estimate client-side training time, using only the measured training time, communicated network speed, and observed dataset size of clients, making it a low-overhead solution which is suitable for real system implementation. We theoretically show the convergence of DTFL on convex and non-convex loss functions under standard assumptions in FL Li et al. (2019); Reisizadeh et al. (2020) and local-loss-based training Belilovsky et al. (2020); Huo et al. (2018); Han et al. (2021). Using DTFL, we train large models (ResNet-56 and ResNet-110 He et al. (2016)) on different number of clients using popular datasets (CIFAR-10 Krizhevsky et al. (2009), CIFAR-100 Krizhevsky et al. (2009), CINIC-10 Darlow et al. (2018), and HAM10000Tschandl et al. (2018)) and their non-I.I.D. (non-identical and independent distribution) variants. We also evaluate the performance of DTFL when employing privacy measures, such as minimizing the distance correlation between raw data and intermediate representations, and shuffling patches of data. The results indicate that DTFL can effectively incorporate privacy techniques without significantly impacting model accuracy. Extensive experimental results show that DTFL can significantly reduce the training time while maintaining model accuracy comparable to state-of-the-art FL methods.

## 2 BACKGROUND AND RELATED WORKS

**Federated Learning.** Existing FL methods (see a comprehensive study of FL Kairouz et al. (2021)) require clients to repeatedly download and update the global model, which is not suitable for training large models with resource-constrained devices in heterogeneous environments and may suffer a severe straggler problem. To address the straggler problem, Li et al. (2019) selects a smaller set of clients for training in each global iteration, but requires more training rounds. Bonawitz et al. (2019) mitigates stragglers by neglecting the slowest 30% clients, while FedProx Li et al. (2020) uses distinct local epoch numbers for clients. Both Bonawitz et al. (2019) and Li et al. (2020) face the challenge of determining the perfect parameters (i.e., percentage of slowest clients and number of local epochs). Recently, tier-based FL methods Chai et al. (2020; 2021); Reisizadeh et al. (2022) propose to divide clients into tiers based on their training speed and select clients from the same tier in each training round to mitigate the straggler problem. However, clients in existing FL methods are required to train the whole global model, which renders significant hurdles in training large models on resource-constrained devices.

**Split Learning.** To tackle the computational limitation of resource-constrained devices, Split Learning (SL) Vepakomma et al. (2018); Gupta & Raskar (2018) splits the global model into a client-side model and a server-side model, and clients need to only update the small client-side model, compared to FL. To increase SL training speed Thapa et al. (2022) incorporated FL into SL, and Wu et al. (2023) proposed a first-parallel-then-sequential approach that clusters clients and sequentially trains a model in SL fashion in each cluster, and then transfers the updated cluster model to the next clusters. In SL, clients must wait for the server's backpropagated gradients to update their models, which can cause significant communication overhead. To address these issues, He et al. (2020a) proposes

FedGKT, to train small models at clients and periodically transfer their knowledge by knowledge distillation to a large server-side model. Han et al. (2021) develops a federated SL algorithm that addresses latency and communication issues by integrating local-loss-based training. Clients train a model using local error signals, which eliminates the need to communicate with the server. However, the client-side models in current SL approaches He et al. (2020a); Han et al. (2021); Zhang et al. (2023) are fixed throughout the training process, and choosing suitable client-side models in heterogeneous environments is challenging as clients' resources may change over time. Compared to these works, the proposed DTFL can dynamically adjust the size of the client-side model for each client over time, which can significantly reduce the training time and mitigate the straggler problem.

## 3 Dynamic Tiering-based Federated Learning

### 3.1 Problem Statement

We aim to collaboratively train a large model (e.g., ResNet, or AlexNet) by $K$ clients on a range of heterogeneous resource-constrained devices that lack powerful computation and communication resources without centralizing the dataset on the server-side. Let $\{(\boldsymbol{x}_i, y_i)\}_{i=1}^{N_k}$ denote the dataset of client $k$, where $\boldsymbol{x}_i$ denotes the $i$th training sample, $y_i$ is the associated label of $\boldsymbol{x}_i$, and $N_k$ is the number of samples in client $k$'s dataset. The FL problem can be formulated as a distributed optimization problem:

$$\min_{\boldsymbol{w}} f(\boldsymbol{w}) \overset{\text{def}}{=} \min_{\boldsymbol{w}} \sum_{k=1}^{K} \frac{N_k}{N} \cdot f_k(\boldsymbol{w}) \quad (1)$$

$$\text{where } f_k(\boldsymbol{w}) = \frac{1}{N_k} \sum_{i=1}^{N_k} \ell\left((\boldsymbol{x}_i, y_i); \boldsymbol{w}\right) \quad (2)$$

where $\boldsymbol{w}$ denotes the model parameters and $N = \sum_{k=1}^{K} N_k$. $f(\boldsymbol{w})$ denotes the global objective function, and $f_k(\boldsymbol{w})$ denotes the $k$th client's local objective function, which evaluates the local loss over its dataset using loss function $\ell$.

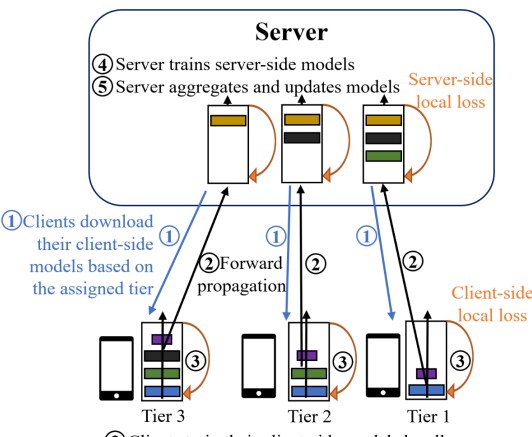

Figure 1: Overview of dynamic tiering-based federated learning. The purple layer at the client-side denotes the auxiliary network in different tiers.

One main drawback of existing federated optimization techniques (e.g., McMahan et al. (2017); Li et al. (2020); Wang et al. (2020b); Reddi et al. (2020)) for solving (1) is that they cannot efficiently train large models on a variety of heterogeneous resource-constrained devices. Such heterogeneity would lead to the severe straggler problem that clients may have significantly different response latencies (i.e., the time between a client receives the training task and returning the results) in the FL process, which would severely slow down the training (see experimental results in Sec. 4.2).

To address these issues, we propose a Dynamic Tiering-based Federated Learning (DTFL) system (see Figure 1), in which we develop a dynamic tier scheduler that assigns clients to suitable tiers based on their training speed. In different tiers, DTFL offloads different portions of the global model to clients and enables each client to update the models in parallel via local-loss-based training, which can reduce the computation and communication demand on resource-constrained devices, while mitigating the straggler problem. Compared with existing works (e.g., He et al. (2020a); Han et al. (2021); Chai et al. (2020)), which can be treated as a single-tier case in DTFL, DTFL provides more flexibility via multiple tiers to cater to a variety of heterogeneous resource-constrained devices in heterogeneous environments. As shown in experimental results in Sec. 4.2, DTFL can significantly reduce the training time while maintaining model accuracy, compared with these methods.

### 3.2 Tiering Local-loss-based Training

To cater for heterogeneous resource-constrained devices, DTFL divides the clients into $M$ tiers based on their training speed. In different tiers, DTFL offloads different portions of the global model $\boldsymbol{w}$ to the server and enables each client to update the models in parallel via local-loss-based training. Specifically, in tier $m$, the model $\boldsymbol{w}$ is split into a client-side model $\boldsymbol{w}^{c_m}$ and a server-side model $\boldsymbol{w}^{s_m}$. Clients in tier $m$ train the client-side model $\boldsymbol{w}^{c_m}$ and an auxiliary network $\boldsymbol{w}^{a_m}$. The auxiliary network is the extra layers connected to the client-side model, and the auxiliary network is

Table 1: Comparison of training time (in seconds) for 10 clients under different tiers when $M = 6$ to achieve 80% accuracy on the I.I.D. CIFAR-10 dataset using ResNet-110. In each experiment, all the clients are assigned to the same tier. Randomly assign clients to different CPU and network speed profiles. Profiles in Case 1: 2 CPUs with 30 Mbps, 1 CPU with 30 Mbps, 0.2 CPU with 30 Mbps. Profiles in Case 2: 4 CPUs with 100 Mbps, 1 CPU with 30 Mbps, 0.1 CPU with 10 Mbps. The experimental setup can be found in Sec. 4.

|  | Tier | 1 | 2 | 3 | 4 | 5 | 6 | FedAvg |
|---|---|---|---|---|---|---|---|---|
| Case1 | Computation Time | 4622 | 8106 | 9982 | 10681 | 11722 | 12250 | 13396 |
| | Communication Time | 5911 | 5995 | 2187 | 2189 | 1018 | 908 | 16 |
| | Overall Training Time | **10533** | 14101 | 12170 | 12871 | 12741 | 13158 | 13408 |
| Case2 | Computation Time | 8384 | 14634 | 17993 | 19027 | 21428 | 22344 | 24428 |
| | Communication Time | 17754 | 18090 | 6720 | 6762 | 2941 | 2653 | 43 |
| | Overall Training Time | 26138 | 32724 | 24713 | 25989 | **24369** | 24997 | 24471 |

used to compute the local loss on the client-side. By introducing the auxiliary network, we enable each client to update the models in parallel with the server Han et al. (2021), which avoids the severe synchronization and substantial communication in SL that significantly slows down the training process Vepakomma et al. (2018); Gupta & Raskar (2018). In this paper, we use a few fully connected layers for the auxiliary network as in Han et al. (2021); Belilovsky et al. (2020); Laskin et al. (2020).

Under this setting, we define $f_k^c(\boldsymbol{w}^{c_m}, \boldsymbol{w}^{a_m})$ as the client-side loss function and $f_k^s(\boldsymbol{w}^{s_m}, \boldsymbol{w}^{c_m})$ as the corresponding server-side loss function in tier $m$. Our goal is to find $\boldsymbol{w}^{c_m*}$ and $\boldsymbol{w}^{a_m*}$ that minimizes the client-side loss function in each tier $m$:

$$\min_{\boldsymbol{w}^{c_m}, \boldsymbol{w}^{a_m}} \sum_{k \in \mathcal{A}^{c_m}} \frac{N_k}{N^m} \cdot f_k^c(\boldsymbol{w}^{c_m}, \boldsymbol{w}^{a_m}) \tag{3}$$

where $f_k^c(\boldsymbol{w}^{c_m}, \boldsymbol{w}^{a_m}) = \frac{1}{N_k} \sum_{i=1}^{N_k} \ell((\boldsymbol{x}_i, y_i); \boldsymbol{w}^{c_m}, \boldsymbol{w}^{a_m})$ and $N^m = \sum_{k \in \mathcal{A}^{c_m}} N_k$. $\mathcal{A}^{c_m}$ denotes the set of clients in tier $m$. Given the optimal client-side model $\boldsymbol{w}^{c_m*}$, the server finds $\boldsymbol{w}^{s_m*}$ that minimizes the server-side loss function:

$$\min_{\boldsymbol{w}^{s_m}} \sum_{k \in \mathcal{A}^{c_m}} \frac{N_k}{N^m} \cdot f_k^s(\boldsymbol{w}^{s_m}, \boldsymbol{w}^{c_m*}) \tag{4}$$

where $f_k^s(\boldsymbol{w}^{s_m}, \boldsymbol{w}^{c_m*}) = \frac{1}{N_k} \sum_{i=1}^{N_k} \ell((\boldsymbol{z}_i, y_i); \boldsymbol{w}^{s_m})$ and $\boldsymbol{z}_i = h_{\boldsymbol{w}^{c_m*}}(\boldsymbol{x}_i)$ is the intermediate output of the client-side model $\boldsymbol{w}^{c_m*}$ given the input $\boldsymbol{x}_i$.

Offloading the model to the server can effectively reduce the total training time, as illustrated in Table 1. As a client offloads more layers to the server (moving towards tier $m = 1$), the model size on the client's side decreases, thereby reducing the computational workload. Meanwhile, this may increase the amount of data transmitted (i.e., the size of the intermediate data and partial model). As indicated in Table 1, there exists a non-trivial tier assignment that minimizes the overall training time. To find the optimal tier assignment, DTFL needs to consider multiple factors, including the communication link speed between the server and the clients, the computation power of each client, and the local dataset size.

### 3.3 DYNAMIC TIER SCHEDULING

In a heterogeneous environment with multiple clients, the proposed dynamic tier scheduling aims to minimize the overall training time by determining the optimal tier assignments for each client.

Specifically, let $m_k^{(r)}$ denote the tier of client $k$ in the training round $r$. $T_k^c(m_k^{(r)})$, $T_k^{com}(m_k^{(r)})$ and $T_k^s(m_k^{(r)})$ represent the training time of the client-side model, the communication time, and the training time of the server-side model of client $k$ at round $r$, respectively. Using the proposed local-loss-based split training algorithm, each client and the server train the model in parallel. The overall training time $T_k$ for client $k$ in each round can be presented as:

$$T_k(m_k^{(r)}) = \max\{T_k^c(m_k^{(r)}) + T_k^{com}(m_k^{(r)}), T_k^s(m_k^{(r)}) + T_k^{com}(m_k^{(r)})\}. \tag{5}$$

As clients train their models in parallel, the overall training time in each round $r$ is determined by the slowest client (i.e., straggler). To minimize the overall training time, we minimize the maximum training time of clients in each round:

$$\min_{\{m_k^{(r)}\}} \max_k T_k(m_k^{(r)}), \text{subject to } \{m_k^{(r)}\} \in \mathbb{M} \ \forall k, \tag{6}$$

where $\mathbb{M}$ denotes the set of tiers. Note that problem (6) is an integer programming problem. To solve (6), it requires the knowledge of each client's training time $\{T_k(m_k^{(r)})\}$ under each tier. As

Table 2: The normalized training times for both client-side and server-side models in different tiers for each client relative to Tier 1, using ResNet-56 with 10 clients. In each experiment, all the clients are assigned to the same tier. We change the CPU capacities of clients in each experiment to evaluate the impact of CPU capacities.

| Tier | 1 | 2 | 3 | 4 | 5 | 6 |
|---|---|---|---|---|---|---|
| Client-side Training Time | $1.00 \pm 0.04$ | $1.63 \pm 0.10$ | $2.16 \pm 0.15$ | $2.68 \pm 0.22$ | $3.30 \pm 0.24$ | $3.81 \pm 0.28$ |
| Server-side Training Time | $1.00 \pm 0.07$ | $0.82 \pm 0.06$ | $0.65 \pm 0.06$ | $0.51 \pm 0.04$ | $0.33 \pm 0.03$ | $0.20 \pm 0.01$ |

the capacities of each client in a heterogeneous environment may change over time, a static tier assignment may still lead to a severe straggler problem. The key question is how to efficiently solve (6) in a heterogeneous environment.

To address this challenge, we develop a **dynamic tier scheduler** to efficiently determine the optimal tier assignments for each client in each round. The idea is to use tier profiling to estimate the training time of each client under each tier, based on which each client will be assigned to the optimal tier.

- **Tier Profiling.** Before the training starts, the server conducts tier profiling to estimate $T_k^c(m_k^{(r)})$, $T_k^{com}(m_k^{(r)})$ and $T_k^s(m_k^{(r)})$ for each client. Specifically, using a standard data batch, the server profiles the transferred data size (i.e., model parameter and intermediate data size) for each tier $m$, as $D_{size}(m_k^{(r)})$. Then, for each client $k$ in tier $m$, the communication time can be estimated as $D_{size}(m_k^{(r)})\tilde{N}_k/\nu_k^{(r)}$, where $\nu_k^{(r)}$ represents the client's communication speed and $\tilde{N}_k$ denotes the number of data batches. To track clients' training time for their respective client-side models, the server maintains and updates the set of historical client-side training times for each client $k$ in tier $m$, denoted as $\mathcal{T}_k^{c_m}$. To mitigate measurement noise, the server uses Exponential Moving Average (EMA) on historical client-side training time (i.e, $\bar{T}_k^{c_m}(m_k^{(r)}) \leftarrow \mathrm{EMA}(\mathcal{T}_k^{c_m}(m_k^{(r)}))$) as the current client's training time in tier $m$. *One main challenge of tier profiling is that to capture the dynamics of the training time of each client in a heterogeneous environment, we need the knowledge of the training time of each client in each tier, but only the training time of each client in the assigned tier is available in each round.* To estimate the training times in other tiers, we study the relationship of the normalized training times among different tiers for each client, where the normalized training time refers to the model training time using a standard data batch. Table 2 shows the normalized training times of different tiers relative to tier 1. As indicated in Table 2, for any client, the normalized training times for both client-side and server-side models in different tiers are the same. *This is because the ratio between the normalized model training times under two different tiers depends on only the model sizes of these two tiers, which does not change if the design of the models under different tiers is given.* Based on this tier profiling, we can estimate the training times in other tiers using the observed training time of each client in the assigned tier (see lines 24 to 29 in Algorithm 1).

- **Tier Scheduling.** In each round, the tier scheduler minimizes the maximum training time of clients. First, it identifies the maximum time (i.e., the straggler training time), denoted as $T_{\max}$, by estimating the maximum training time of all clients if they are assigned to a tier that minimizes their training time (see line 31 in Algorithm 1). Then, it assigns other clients to a tier with an estimated training time that is less than or equal to $T_{\max}$ (see line 33 in Algorithm 1). To better utilize the resources of each client, the tier scheduler selects tier $m$ that minimizes the offloading to the server while still ensuring that its estimated training time remains below $T_{\max}$ by $m_k^{(r+1)} \leftarrow \arg\max_m \left( \{\hat{T}_k(m_k^{(r)}) \leq T_{\max}\} \right)$.

The dynamic tier scheduler is detailed in **TierScheduler**($\cdot$) function in Algorithm 1. The DTFL training process (illustrated in Figure 1) is described in in Algorithm 1.

### 3.4 CONVERGENCE ANALYSIS

We show the convergence of both client-side and server-side models in DTFL on convex and non-convex loss functions based on standard assumptions in FL and local-loss-based training. We assume that **(A1)** client-side $f_k^{c_m}$ and server-side $f_k^{s_m}$ objective functions of each client in each tier are differentiable and $L$-smooth; **(A2)** $f_k^{c_m}$ and $f_k^{s_m}$ have expected squared norm bounded by $G_1^2$; **(A3)** the variance of the gradients of $f_k^{c_m}$ and $f_k^{s_m}$ is bounded by $\sigma^2$; **(A4)** $f_k^{c_m}$ and $f_k^{s_m}$ are $\mu$-convex for $\mu \geq 0$ for some results; **(A5)** the client-side objective functions are $(G_2, B)$-BGD (Bounded

---

**Algorithm 1 DTFL's Training Process,**

**Initialization**
**MainServer()**
1: **for** each round $r = 0$ to $R - 1$ **do**
2:     $\boldsymbol{m}^{(r)} \leftarrow \textbf{TierScheduler}(\boldsymbol{T}^{c_m}(m_k^{(r)}), \boldsymbol{\nu}^{(r)}, \tilde{\boldsymbol{N}})$
3:     **for** each client $k$ in parallel **do**
4:         $(\boldsymbol{z}_k^{(r)}, \mathbf{y}_k) \leftarrow \textbf{ClientUpdate}(\boldsymbol{w}_k^{c_m^{(r)}})$
5:         Measure $T_k^{c_m}(m_k^{(r)})$, $\nu_k^{(r)}$ and $\tilde{N}_k$
         //server updates the server-side model
6:         Forward propagation of $\boldsymbol{z}_k^{(r)}$ on $\boldsymbol{w}_k^{s_m^{(r)}}$
7:         Calculate loss, back propagation on $\boldsymbol{w}_k^{s_m^{(r)}}$
8:         $\boldsymbol{w}_k^{s_m^{(r+1)}} \leftarrow \boldsymbol{w}_k^{s_m^{(r)}} - \eta \nabla f_k^s(\boldsymbol{w}^{s_m}, \boldsymbol{w}^{c_m *})$
9:         Receive updated $\boldsymbol{w}_k^{c_m^{(r+1)}}$ from client $k$
10:         $\boldsymbol{w}_k^{(r+1)} = \{\boldsymbol{w}_k^{c_m^{(r+1)}}, \boldsymbol{w}_k^{s_m^{(r+1)}}\}$
11:     **end for**
12:     $\boldsymbol{w}^{(r+1)} = \frac{1}{K} \sum_k \boldsymbol{w}_k^{(r+1)}$
13:     Update all models ($\boldsymbol{w}^{c_m^{(r+1)}}$ and $\boldsymbol{w}^{s_m^{(r+1)}}$) in each tier using $\boldsymbol{w}^{(r+1)}$
14: **end for**

**ClientUpdate($\boldsymbol{w}_k^{c_m^{(r)}}$)**
15: Forward propagate on local data to calculate $\boldsymbol{z}_k^{(r)}$
16: Send $(\boldsymbol{z}_k^{(r)}, y_k)$ to the server
17: Forward propagation to the auxiliary layer
18: Calculate local loss, back propagation

19: $\boldsymbol{w}_k^{c_m^{(r+1)}} \leftarrow \boldsymbol{w}_k^{c_m^{(r)}} - \eta \nabla f_k^{c_m}(\boldsymbol{w}^{c_m^{(r)}}, \boldsymbol{w}^{a_m^{(r)}})$
20: Send $\boldsymbol{w}_k^{c_m^{(r+1)}}$ to the server

**TierScheduler($\boldsymbol{T}^{c_m}(m_k^{(r)}), \boldsymbol{\nu}^{(r)}, \tilde{\boldsymbol{N}}$)**
21: **for** all client $k$ **do**
22:     Add $\left( T_k^{c_m}(m_k^{(r)}) - \frac{D^m \tilde{N}_k}{\nu_k^{(r)}} \right)$ into $\mathcal{T}_k^{c_m}(m_k^{(r)})$
23:     $\bar{T}_k^{c_m}(m_k^{(r)}) \leftarrow \text{EMA}\left( \mathcal{T}_k^{c_m}(m_k^{(r)}) \right)$
24:     **for** all tier $m_k^{(r+1)}$ **do**
         //estimate $\hat{T}_k(m_k^{(r+1)})$
25:         $\hat{T}_k^{com}(m_k^{(r+1)}) \leftarrow \frac{D_{size}(m_k^{(r)})\tilde{N}_k}{\nu_k^{(r)}}$
26:         $\hat{T}_k^c(m_k^{(r+1)}) \leftarrow \frac{T^{c_p}(m_k^{(r+1)})}{T^{c_p}(m_k^{(r)})}\bar{T}_k^{c_m}(m_k^{(r)})$
27:         $\hat{T}_k^s(m_k^{(r+1)}) \leftarrow T^{s_p}(m_k^{(r+1)})\tilde{N}_k$
28:         Compute $\hat{T}_k(m_k^{(r+1)})$ using Equation (5)
29:     **end for**
30: **end for**
31: $T_{\max} \leftarrow \max\limits_k \min\limits_m \{\hat{T}_k(m_k^{(r+1)})\}$
32: **for** all clients $k$ **do**
33:     $m_k^{(r+1)} \leftarrow \arg\max\limits_m \left( \{\hat{T}_k(m_k^{(r+1)}) \leq T_{\max}\} \right)$
34: **end for**
35: **Return** $\boldsymbol{m}^{(r+1)}$

---

Gradient Dissimilarity); **(A6)** the time-varying parameter satisfies $d^{c_m^{(r)}} < \infty$. These assumptions are well-established and frequently utilized in the machine learning literature for convergence analyses, as in previous works such as Stich (2018); Li et al. (2019); Belilovsky et al. (2020); Yu et al. (2019); Karimireddy et al. (2020). We adopt the approach of Belilovsky et al. (2020) for local-loss-based training, where the server input distribution varies over time and depends on client-side model convergence.

**Theorem 1 (Convergence of DTFL)** *Under assumptions (A1), (A2), (A3), and (A5), the convergence properties of DTFL for both convex and non-convex functions are summarized as follows:* **Convex:** *Under (A4), $\eta \leq \frac{1}{8L(1+B^2)}$ and $R \geq \frac{4L(1+B^2)}{\mu}$, the client-side model converges at the rate of $\mathcal{O}\left(\mu D^2 \exp\left(-\frac{\eta}{2}\mu R\right) + \frac{\eta H_1^2}{\mu R A^m}\right)$ and the server-side model converges at the rate of $\mathcal{O}\left(\frac{C_1}{R} + \frac{H_2\sqrt{F^{s_m^0}}}{\sqrt{R A^m}} + \frac{F^{s_m^0}}{\eta_{max} R}\right)$.* **Non-convex:** *If both $f^{c_m}$ and $f^{s_m}$ are non-convex with $\eta \leq \frac{1}{8L(1+B^2)}$, then the client-side model converges at the rate of $\mathcal{O}\left(\frac{H_1\sqrt{F^{c_m^0}}}{\sqrt{R A^m}} + \frac{F^{c_m^0}}{\eta_{max} R}\right)$ and the server-side model converges at the rate of $\mathcal{O}\left(\frac{C_2}{R} + \frac{H_2\sqrt{F^{s_m^0}}}{\sqrt{R A^m}} + \frac{F^{s_m^0}}{\eta_{max} R}\right)$, where $\eta_{max}$ is the maximum of learning rate $\eta$, $H_1^2 := \sigma^2 + \left(1 - \frac{A^m}{K}\right)G_2^2$, $H_2^2 := L^3\left(B^2 + 1\right)F^{s_m} + \left(1 - \frac{A^m}{K}\right)L^2 G_2^2$, $D := \left\|\boldsymbol{w}^{c_m^0} - \boldsymbol{w}^{c_m^\star}\right\|$, $F^{c_m^0} := f^{c_m}\left(\boldsymbol{w}^{c_m^0}\right)$, and $F^{s_m^0} := f^{s_m}\left(\boldsymbol{w}^{s_m^0}\right)$. $C_1 = G_1\sqrt{G_2^2 + 2LB^2 F^{s_m^0} \sum_r d^{c_m^{(r)}}}$ and $C_2 = G_1\sqrt{G_2^2 + B^2 G_1^2 \sum_r d^{c_m^{(r)}}}$ are convergent based on (A6). $A^m = \min_r\{A^{c_m^{(r)}} > 0\}$, where $A^{c_m^{(r)}}$ denotes the number of clients in tier $m$ at round $r$. $d^{c_m^{(r)}}$ denotes the distance between the density function of the output of the client-side model and its converged state.*

According to Theorem 1, both client-side and server-side models converge as the number of rounds $R$ increases, with varying convergence rates across different tiers. Note that as DTFL leverages

the local-loss-based split training, the convergence of the server-side model depends on the convergence of the client-side model, which is explicitly characterized by $C_1$ and $C_2$ in the analysis. The complete proof of the theorem is given in Appendix B.

## 4 EXPERIMENTAL EVALUATION

### 4.1 EXPERIMENTAL SETUP

**Dataset.** We consider image classification on four public image datasets, including CIFAR-10 Krizhevsky et al. (2009), CIFAR-100 Krizhevsky et al. (2009), CINIC-10 Darlow et al. (2018), and HAM10000Tschandl et al. (2018). We also consider label distribution skew Li et al. (2022) (i.e., the distribution of labels varies across clients) to generate their non-I.I.D. variants using He et al. (2020b). Appendix A describes the dataset distributions used in these experiments.

**Baselines.** We compare DTFL with state-of-the-art FL/SL methods, including FedAvg McMahan et al. (2017), SplitFed Thapa et al. (2022), FedYogi Reddi et al. (2020), and FedGKT He et al. (2020a). For the same reasons as in He et al. (2020a), we do not compare with FedProx Li et al. (2020) and FedMA Wang et al. (2020a). FedProx Li et al. (2020) performs worse than FedAvg in the large convolutional neural networks, CNN, setting and FedMA cannot work on modern DNNs that contain batch normalization layers (e.g., ResNet).

**Implementation.** We conducted the experiment using Python 3.11.3 and the PyTorch library version 1.13.1, which is available online in Anonymous (2023). The DTFL and the baselines are deployed in a server, which is equipped with dual-sockets Intel(R) Xeon(R) CPU E5-2630 v4 @ 2.20GHz with hyper-threading disabled, and four NVIDIA GeForce GTX 1080 Ti GPUs, 64 GB of memory. Each client is assigned a different simulated CPU and communication resource in order to simulate heterogeneous resources (i.e., simulate the training time of different CPU/network profiles). By using these resource profiles, we simulate a heterogeneous environment where clients' capacity varies in both cross-solo and cross-device FL settings. We consider 5 resource profiles: 4 CPUs with 100 Mbps, 2 CPUs with 30 Mbps, 1 CPU with 30 Mbps, 0.2 CPU with 30 Mbps, and 0.1 CPU with 10 Mbps communication speed to the server. Each client is assigned one resource profile at the beginning of the training, and the profile can be changed during the training process to simulate the dynamic environment.

**Model Architecture.** DTFL is a versatile approach suitable for training a wide range of neural network models (e.g., Multilayer Perceptron, MLP, Recurrent Neural Networks, RNN, and CNN), particularly benefiting large-scale models. In the experiments, we evaluate large CNN models, ResNet-56 and ResNet-110 He et al. (2016) that work well on the selected datasets. Furthermore, DTFL can also be applied to large language models (LLM) like BERT Devlin et al. (2018) by splitting techniques as proposed in Tian et al. (2022); Lit et al. (2022). For each tier, we split a global model to create client and server-side models. The split layer is different in tiers, and it moves toward the last layer as the tier increases. For each client-side model, we add a fully connected (f.c.) and an average pooling (avgpool) layer as the auxiliary network. More details can be found in Appendix A.5. We follow the same setting as He et al. (2020a) for FedGKT. We split the global model after module 2 (as defined in Appendix A.5) for the SplitFed model.

### 4.2 TRAINING TIME IMPROVEMENT OF DTFL

**Training time comparison of DTFL to baselines.** In Table 3, we summarize all experimental results of training a global model (i.e., ResNet-56 or ResNet-110) with 7 tiers (i.e., $M = 7$) when using different federated learning methods. The experiments were conducted on a heterogeneous client population, with 20% assigned to each profile at the experiment's outset. Every 50 rounds, the client profiles (i.e., number of simulated CPUs and communication speed) of 30% of the clients were randomly changed to simulate a dynamic environment, while all clients participated in every training round. The training time of each method to achieve a target accuracy is provided in Table 3. In all cases for both I.I.D. and non-I.I.D. settings, DTFL significantly reduces the training time, compared to baselines (FedAvg, SplitFed, FedYogi, FedGKT). For example, DTFL reduces the training time of FedAvg by 80% to reach the target accuracy on I.I.D. CIFAR-10 with ResNet-110. This experiment illustrates the capabilities of DTFL which can significantly reduce training time when training on distributed heterogeneous clients. Figure 2 illustrates the curve of the test accuracy during the training process of all the methods for the I.I.D. CIFAR-10 case with ResNet-110, where we observe a faster convergence using DTFL, compared with baselines.

Table 3: Comparison of training time (in seconds) to baseline approaches with 10 clients on different datasets. The numbers represent the training time used to achieve the target accuracy (i.e., CIFAR-10 I.I.D. 80%, CIFAR-10 non-I.I.D. 70%, CIFAR-100 I.I.D. 55%, CIFAR-100 non-I.I.D. 50%, CINIC-10 I.I.D. 75%, CINIC-10 non-I.I.D. 65%, and HAM10000 75%).

| Method | Global Model | CIFAR-10 I.I.D. | CIFAR-10 non-I.I.D. | CIFAR-100 I.I.D. | CIFAR-100 non-I.I.D. | CINIC-10 I.I.D. | CINIC-10 non-I.I.D. | HAM10000 |
|---|---|---|---|---|---|---|---|---|
| DTFL | ResNet-56 | **2750** | **3986** | **3585** | **6093** | **23968** | **40138** | **2353** |
|  | ResNet-110 | **4816** | **7054** | **5678** | **9874** | **42099** | **70469** | **3615** |
| FedAvg | ResNet-56 | 13157 | 20773 | 19170 | 35350 | 114509 | 197926 | 11566 |
|  | ResNet-110 | 24471 | 39094 | 36360 | 66317 | 210468 | 395423 | 22328 |
| SplitFed | ResNet-56 | 35877 | 46514 | 54174 | 97859 | 271873 | 510156 | 19549 |
|  | ResNet-110 | 67265 | 84342 | 101783 | 183122 | 521334 | 896627 | 43581 |
| FedYogi | ResNet-56 | 9122 | 13130 | 12727 | 19216 | 82083 | 113464 | 8071 |
|  | ResNet-110 | 19299 | 25668 | 23978 | 35356 | 155212 | 219134 | 14932 |
| FedGKT | ResNet-56 | 25458 | 30808 | 36838 | 59461 | 184589 | 218065 | 37181 |
|  | ResNet-110 | 39676 | 47458 | 64457 | 98754 | 321534 | 411259 | 61755 |

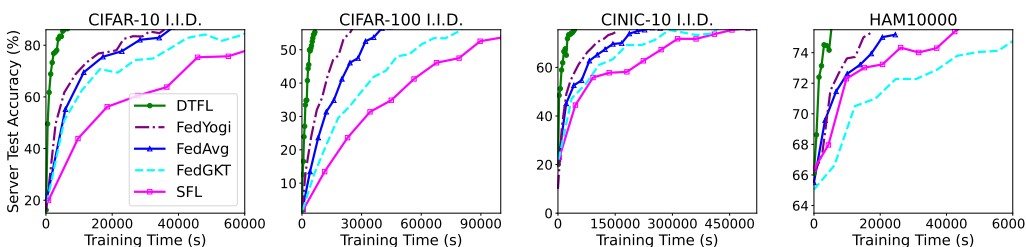

Figure 2: Comparing the training process of DTFL with baselines for the I.I.D. CIFAR-10 dataset.

### 4.3 UNDERSTANDING DTFL UNDER DIFFERENT SETTINGS

**Performance of DTFL with different numbers of clients.** We evaluate the performance of DTFL with different numbers of clients to better understand the scalability of DTFL. Table 4 shows the training time for various training methods using different numbers of clients on the I.I.D. CIFAR-10 dataset, to reach a target accuracy 80% with the ResNet-110 model. In these experiments, we randomly sampled 10% of all clients to be involved in each round of the training process. Note that DTFL can also be employed in other FL client selection methods (e.g., Chai et al. (2020; 2021)). In general, increasing the number of clients has no adverse effects on DTFL performance and significantly reduces training time compared to other methods.

Table 4: Performance of DTFL with different numbers of clients.

| # Clients | DTFL | FedAvg | SplitFed | FedYogi | FedGKT |
|---|---|---|---|---|---|
| 20 | **1877** | 7950 | 21350 | 6341 | 14595 |
| 50 | **2547** | 10435 | 29026 | 8073 | 17872 |
| 100 | **3102** | 14032 | 36449 | 10760 | 24438 |
| 200 | **3594** | 16060 | 43942 | 12786 | 27632 |

**Impact of the number of tiers on DTFL performance.** We evaluate the DTFL performance under different numbers of tiers while employing the global ResNet-110 model (model details under different tiers are provided in Table 11 in the appendix). In Figure 3, we present the total training time for the I.I.D. CIFAR-10 dataset and 10 clients with different numbers of tiers. We conducted experiments with two different cases, similar to those in Table 1, where clients' CPU profiles randomly switch to another profile every 20 rounds of training within the profiles of the same case. Experiments show that to reach the target accuracy of 80%, the training

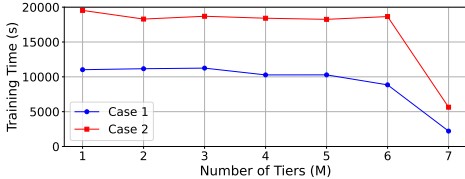

Figure 3: Impact of the number of tiers on the total training time.

time generally decreases with the number of tiers, as DTFL would have more flexibility to fine-tune the tier of each client based on the heterogeneous resources of each client. It should be noted that the model under each tier needs to be carefully designed based on the structure of the global model. A client-side model obtained by arbitrarily splitting the global model may negatively impact the model accuracy. Thus, the maximum suitable number of tiers is much less than the number of layers of a global model. For ResNet-110, we find 7 tiers provided in Table 11 in the appendix can significantly reduce the training time while maintaining the model accuracy.

### 4.4 PRIVACY DISCUSSION

Using DTFL, we can significantly reduce the training time. However, exchanging hidden feature maps (i.e., the intermediate output $z_i$) may potentially leak privacy. A potential threat to DTFL is model inversion attacks, extracting client data by analyzing feature maps or model parameter transfers from clients to servers. Prior research Yin et al. (2021); Zhu et al. (2019) has shown that attackers need access to all model parameters or gradients to recover client data. This is not feasible from partial or fragmented models. Thus, similar to Thapa et al. (2022), DTFL can use separate servers for model aggregation and training to prevent a single server from having access to all model parameters and intermediate data. Another potential threat to DTFL is that an attacker can infer client model parameters by inputting dummy data into the client's local model and training a replicating model on the resulting feature maps Shen et al. (2023). DTFL can prevent this attack by denying clients access to external datasets, query services, and dummy data, thereby preventing the attacker from obtaining the necessary data.

However, for attackers with strong eavesdropping capabilities, there may be potential privacy leakage. As DTFL is compatible with privacy-preserving federated learning approaches, existing data privacy protection methods can be easily integrated into DTFL to mitigate potential privacy leakage, e.g., distance correlation Vepakomma et al. (2020), differential privacy Abadi et al. (2016), patch shuffling Yao et al. (2022), PixelDP Lecuyer et al. (2019), SplitGuard Erdogan et al. (2022), and cryptography techniques Sami & Güler (2023); Qiu et al. (2023). For example, we can add a regularization term into the client's local training objective to reduce the mutual information between hidden feature maps and raw data Wang et al. (2021), making it more difficult for attackers to reconstruct raw data. Each client decorrelates its input $x_i$ and related feature map $z_i$, i.e., $f_k^{c,private}(\boldsymbol{w}^{c_m}, \boldsymbol{w}^{a_m}) = (1-\alpha)f_k^c(\boldsymbol{w}^{c_m}, \boldsymbol{w}^{a_m}) + \alpha DCor(\mathbf{x_i}, \mathbf{z_i})$, where $\alpha$ balances the model performance and the data privacy, and $DCor$ denotes the distance correlation defined in Vepakomma et al. (2020). Distance correlation enhances the privacy of DTFL against reconstruction attacks Vepakomma et al. (2020).

**Integration of privacy protection methods.** We evaluate the model accuracy and privacy trade-offs of DTFL when integrating distance correlation and patch shuffling techniques. Table 5 illustrates the model accuracy of DTFL with distance correlation, showing a decreasing trend as $\alpha$ increases. This suggests that integrating distance correlation can enhance data privacy without significant accu-

Table 5: Impact of integrating privacy protection into DTFL on the CIFAR-10 dataset using ResNet-56 with 20 clients.

| Method | Distance Correlation ($\alpha$) | | | | Patch Shuffling |
|---|---|---|---|---|---|
| | 0.00 | 0.25 | 0.50 | 0.75 | |
| Accuracy | 87.1 | 86.8 | 83.5 | 75.6 | 85.4 |

racy loss, especially for relatively smaller values of $\alpha$. Notably, applying patch shuffling with the same settings as in Yao et al. (2022) to intermediate data has minimal impact on accuracy. The server lacks information about the clients' $\alpha$ values, which can vary between clients. This prevents the server from inferring the data of the clients.

## 5 CONCLUSION

In this paper, we developed DTFL as an effective solution to address the challenges of training large models collaboratively in a heterogeneous environment. DTFL offloads different portions of the global model to clients in different tiers and allows each client to update the models in parallel using local-loss-based training, which can meet computation and communication requirements on resource-constrained devices and mitigate the straggler problem. We developed a dynamic tier scheduling algorithm, which dynamically assigns clients to appropriate tiers based on their training time. The convergence of DTFL is analyzed theoretically. Extensive experiments on large datasets with different numbers of highly heterogeneous clients show that DTFL can significantly reduce the training time while maintaining model accuracy, compared with state-of-the-art FL methods.

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

# APPENDIX

## A    MORE DETAILS ABOUT EXPERIMENTS

Table 6 summarizes the notations used in this paper.

Table 6: Summary of notations used in the paper

| Symbol | Description |
|---|---|
| $K, k$ | Number and index of clients |
| $\boldsymbol{x}_i, y_i$ | $i$th training sample and associated label |
| $N_k, N$ | Size of $k$th training dataset and total training dataset size |
| $R, r$ | Number and index of global rounds |
| $A^{c_m^{(r)}}, \mathcal{A}^{c_m^{(r)}}$ | Number and Set of clients in tier $m$, at round $r$ |
| $\boldsymbol{x}_n, y_n, n$ | $n$th training sample, $n$th label, index of datapoint |
| $\boldsymbol{w}$ | Model parameters |
| $d, p$ | Distance to converged output of client-side model, probability distribution |
| $D$ | Distance of the model to the optimal model |
| $D_{size}(\cdot)$ | Size of data transferred (MB) using profiling |
| $m_k^{(r)}, M, \mathbb{M}$ | Tier of client $k$ at round $r$, number, and set of tiers |
| $\nu$ | Communication speed |

### A.1    DATASET

As this paper considers training large models, we do not use the LEAF benchmark Caldas et al. (2018) datasets because the benchmark datasets offered are either very small or the datasets they contain are too simple for large convolutional neural networks (CNN) which cannot suitably evaluate our algorithm when running on large CNN models.

As the performance of different models is affected by the data distribution in a non-I.I.D. setting, we fixed the distribution of the non-I.I.D. dataset (Dirichlet distribution with a concentration parameter of 0.5) with a fixed random seed for a fair comparison. Table 7 describes the non-I.I.D. distribution that is used in the experiments with 10 clients.

### A.2    NUMBER OF TIERS.

DTFL is adaptable to diverse client dynamics and can be applied to various neural network models. In our experiments with ResNet-56 and ResNet-110, we examine different tier configurations, focusing on a 7-tier setup ($M = 7$) based on our models and client profiles.

### A.3    HYPER-PARAMETERS.

We tuned hyperparameters to the dataset and used the same hyperparameters for the client and server sides in each experiment. ADAM optimizer is selected for all datasets and their variations. We set the initial learning rate $\eta_0$ as 0.001 for CIFAR-10, CIFAR-100, CINIC-10, and 0.0001 for HAM10000. Once the accuracy has reached a plateau the learning rate is reduced by a factor of 0.9. The local batch size of each client is 50 when there are 200 clients and in all other experiments is 100. The local epoch is 1 for all experiments.

### A.4    HETEROGENEOUS DATA DISTRIBUTION (NON-I.I.D.)

We have observed that DTFL outperforms baselines in various non-I.I.D. distributions. To ensure fair comparisons across different approaches, we adopt a consistent non-I.I.D. distribution. Specifically, we employ a Dirichlet distribution with a concentration parameter of 0.5 and a fixed random seed for experiments involving non-I.I.D. datasets. For instance, you can refer to Table 7 for details regarding the non-I.I.D. distribution used in experiments with 10 clients.

### A.5    MODEL ARCHITECTURE

ResNet-56 and ResNet-110 are large Residual Networks that we used as the global model in our experiments. We define modules of the model as part of a model that contains some adjacent layers.

Table 7: The heterogeneous data label distribution (non-I.I.D.) for CIFAR-10

| Client ID | Numbers of Samples in Each Class | | | | | | | | | | Total |
|---|---|---|---|---|---|---|---|---|---|---|---|
| | $c_0$ | $c_1$ | $c_2$ | $c_3$ | $c_4$ | $c_5$ | $c_6$ | $c_7$ | $c_8$ | $c_9$ | |
| k = 0 | 372 | 2398 | 518 | 2 | 1036 | 641 | 210 | 0 | 0 | 0 | 5177 |
| k = 1 | 191 | 84 | 77 | 1008 | 917 | 305 | 0 | 263 | 1295 | 736 | 4876 |
| k = 2 | 23 | 362 | 1281 | 40 | 358 | 1011 | 123 | 451 | 316 | 284 | 4249 |
| k = 3 | 97 | 1032 | 289 | 185 | 670 | 0 | 178 | 84 | 1048 | 1467 | 5050 |
| k = 4 | 40 | 130 | 1209 | 186 | 5 | 57 | 3307 | 1176 | 0 | 0 | 6110 |
| k = 5 | 1639 | 0 | 296 | 121 | 68 | 717 | 403 | 372 | 1932 | 0 | 5548 |
| k = 6 | 1 | 866 | 60 | 101 | 451 | 598 | 120 | 83 | 323 | 2316 | 4919 |
| k = 7 | 1307 | 15 | 428 | 0 | 290 | 2 | 356 | 1448 | 50 | 192 | 4088 |
| k = 8 | 849 | 0 | 88 | 910 | 1187 | 1414 | 24 | 229 | 36 | 4 | 4741 |
| k = 9 | 481 | 113 | 754 | 2447 | 18 | 225 | 279 | 894 | 0 | 1 | 5242 |

Then for each tier, we split a global model between the client-side and the server-side based on these modules. Table 8 and 9 show details of the ResNet-56 and ResNet-110 models and how we split these models into 8 modules. For each client-side model, we add a fully connected (f.c.) and an average pooling (avgpool) layer as the auxiliary network. The input dimension of the f.c. layer is adjusted to match the output of each client-side model. Table 10 shows the model architecture of each tier for the different number of modules ($m$) used in this paper.

We follow the same setting as He et al. (2020a) for FedGKT. We split the global model after module $md2$ for SplitFed.

### A.6 DIFFERENT NUMBER OF TIERS

Table 11 shows how the model is split between client and server as the number of tiers changes.

### A.7 THE DETAILED TRAINING PROCESS OF DTFL

The training process of DTFL in each round is described in the following steps, which are detailed in Algorithm 1 and illustrated in Figure 1.

① **Model download.** In round $r$, the dynamic tier scheduler assigns each client $k$ to an appropriate tier $m_k^{(r)}$ using **TierScheduler**($\cdot$) function. Then each client downloads the client-side model $w_k^{c_m^{(r)}}$ from the server.

② **Local forward propagation.** Each client performs forward propagation in parallel using its local data on the downloaded model $w_k^{c_m^{(r)}}$ and passes the intermediate data $z_k^{(r)}$ and the corresponding label $y_k$ to the server.

③ **Local training and update.** Based on the local loss, each client $k$ updates its model $w_k^{c_m^{(r)}}$ (i.e., $w_k^{c_m^{(r+1)}} \leftarrow w_k^{c_m^{(r)}} - \eta \nabla f_k^{c_m}(w^{c_m^{(r)}}, w^{a_m^{(r)}})$).

④ **Server-side training and update, which runs in parallel with step ③.** The server continues the forward-propagation and back-propagation on the server-side model $w_k^{s_m^{(r)}}$ for each client $k$ in parallel. Then server updates the server-side model $w_k^{s_m^{(r)}}$ (i.e., $w_k^{s_m^{(r+1)}} \leftarrow w_k^{s_m^{(r)}} - \eta \nabla f_k^s(w^{s_m^{(r)}}, w^{c_m*})$).

⑤ **Global model update and aggregation.** At the final step of each round $r$, the server first aggregates the client-side and the server-side models for each client (i.e., $w_k^{(r+1)} = \{w_k^{c_m^{(r+1)}}, w_k^{s_m^{(r+1)}}\}$). Then the server updates the global model by averaging all the models $w_k^{(r+1)}$, (i.e., $w^{(r+1)} = \frac{1}{N} \sum_k w_k^{(r+1)}$). Based on $w^{(r+1)}$, the server updates all the models ($w^{c_m^{(r+1)}}$ and $w^{s_m^{(r+1)}}$) in each tier.

Table 8: Detailed architecture of the ResNet-56 used in our experiment

| Module | Parameter & Shape (cin, cout, kernal size) | # |
|---|---|---|
| md1 | conv1: $3 \times 16 \times 3 \times 3$, stride:(1,1); padding:(1,1) 
 maxpool: $3 \times 1$ | $\times 1$ |
| md2 – | conv1: $16 \times 16 \times 1 \times 1$, stride: $(1,1)$ 
 conv2: $16 \times 16 \times 3 \times 3$, stride: $(1,1)$; padding: $(1,1)$ 
 conv3: $16 \times 64 \times 1 \times 1$, stride: $(1,1)$ 
 downsample.conv: $16 \times 64 \times 1 \times 1$, stride: $(1,1)$ | $\times 1$ |
| | conv1: $64 \times 16 \times 1 \times 1$, stride: $(1,1)$ 
 conv2: $16 \times 16 \times 3 \times 3$, stride: $(1,1)$; padding: $(1,1)$ 
 conv3: $16 \times 64 \times 1 \times 1$, stride: $(1,1)$ | $\times 2$ |
| md3 | conv1: $64 \times 16 \times 1 \times 1$, stride: $(1,1)$ 
 conv2: $16 \times 16 \times 3 \times 3$, stride: $(1,1)$; padding: $(1,1)$ 
 conv3: $16 \times 64 \times 1 \times 1$, stride: $(1,1)$ | $\times 3$ |
| md4 – | conv1: $64 \times 32 \times 1 \times 1$, stride: $(1,1)$ 
 conv2: $32 \times 32 \times 3 \times 3$, stride: $(1,1)$; padding: $(1,1)$ 
 conv3: $32 \times 128 \times 1 \times 1$, stride: $(1,1)$ 
 downsample.conv: $64 \times 128 \times 1 \times 1$, stride: $(2,2)$ | $\times 1$ |
| | conv1: $128 \times 32 \times 1 \times 1$, stride: $(1,1)$ 
 conv2: $32 \times 32 \times 3 \times 3$, stride: $(1,1)$; padding: $(1,1)$ 
 conv3: $32 \times 128 \times 1 \times 1$, stride: $(1,1)$ | $\times 2$ |
| md5 | conv1: $128 \times 32 \times 1 \times 1$, stride: $(1,1)$ 
 conv2: $32 \times 32 \times 3 \times 3$, stride: $(1,1)$; padding: $(1,1)$ 
 conv3: $32 \times 128 \times 1 \times 1$, stride: $(1,1)$ | $\times 3$ |
| md6 – | conv1: $128 \times 64 \times 1 \times 1$, stride: $(1,1)$ 
 conv2: $64 \times 64 \times 3 \times 3$, stride: $(1,1)$; padding: $(1,1)$ 
 conv3: $64 \times 256 \times 1 \times 1$, stride: $(1,1)$ 
 downsample.conv: $128 \times 256 \times 1 \times 1$, stride: $(2,2)$ | $\times 1$ |
| | conv1: $256 \times 64 \times 1 \times 1$, stride: $(1,1)$ 
 conv2: $64 \times 64 \times 3 \times 3$, stride: $(1,1)$; padding: $(1,1)$ 
 conv3: $64 \times 256 \times 1 \times 1$, stride: $(1,1)$ | $\times 2$ |
| md7 | conv1: $256 \times 64 \times 1 \times 1$, stride: $(1,1)$ 
 conv2: $64 \times 64 \times 3 \times 3$, stride: $(1,1)$; padding: $(1,1)$ 
 conv3: $64 \times 256 \times 1 \times 1$, stride: $(1,1)$ | $\times 3$ |
| md8 | avgpool 
 fc: $256 \times 10$ | $\times 1$ 
 $\times 1$ |

Steps ① to ⑤ are repeated in each round to train a global model.

Table 9: Detailed architecture of the ResNet-110 used in our experiment

| Module | Parameter & Shape (cin, cout, kernal size) | # |
|---|---|---|
| md1 | conv1: $3 \times 16 \times 3 \times 3$, stride:(1,1); padding:(1,1)
maxpool: $3 \times 1$ | $\times 1$ |
| md2
–￼ | conv1: $16 \times 16 \times 1 \times 1$, stride: $(1,1)$
conv2: $16 \times 16 \times 3 \times 3$, stride: $(1,1)$; padding: $(1,1)$
conv3: $16 \times 64 \times 1 \times 1$, stride: $(1,1)$
downsample.conv: $16 \times 64 \times 1 \times 1$, stride: $(1,1)$ | $\times 1$ |
| | conv1: $64 \times 16 \times 1 \times 1$, stride: $(1,1)$
conv2: $16 \times 16 \times 3 \times 3$, stride: $(1,1)$; padding: $(1,1)$
conv3: $16 \times 64 \times 1 \times 1$, stride: $(1,1)$ | $\times 5$ |
| md3 | conv1: $64 \times 16 \times 1 \times 1$, stride: $(1,1)$
conv2: $16 \times 16 \times 3 \times 3$, stride: $(1,1)$; padding: $(1,1)$
conv3: $16 \times 64 \times 1 \times 1$, stride: $(1,1)$ | $\times 6$ |
| md4
– | conv1: $64 \times 32 \times 1 \times 1$, stride: $(1,1)$
conv2: $32 \times 32 \times 3 \times 3$, stride: $(1,1)$; padding: $(1,1)$
conv3: $32 \times 128 \times 1 \times 1$, stride: $(1,1)$
downsample.conv: $64 \times 128 \times 1 \times 1$, stride: $(2,2)$ | $\times 1$ |
| | conv1: $128 \times 32 \times 1 \times 1$, stride: $(1,1)$
conv2: $32 \times 32 \times 3 \times 3$, stride: $(1,1)$; padding: $(1,1)$
conv3: $32 \times 128 \times 1 \times 1$, stride: $(1,1)$ | $\times 5$ |
| md5 | conv1: $128 \times 32 \times 1 \times 1$, stride: $(1,1)$
conv2: $32 \times 32 \times 3 \times 3$, stride: $(1,1)$; padding: $(1,1)$
conv3: $32 \times 128 \times 1 \times 1$, stride: $(1,1)$ | $\times 6$ |
| md6
– | conv1: $128 \times 64 \times 1 \times 1$, stride: $(1,1)$
conv2: $64 \times 64 \times 3 \times 3$, stride: $(1,1)$; padding: $(1,1)$
conv3: $64 \times 256 \times 1 \times 1$, stride: $(1,1)$
downsample.conv: $128 \times 256 \times 1 \times 1$, stride: $(2,2)$ | $\times 1$ |
| | conv1: $256 \times 64 \times 1 \times 1$, stride: $(1,1)$
conv2: $64 \times 64 \times 3 \times 3$, stride: $(1,1)$; padding: $(1,1)$
conv3: $64 \times 256 \times 1 \times 1$, stride: $(1,1)$ | $\times 5$ |
| md7 | conv1: $256 \times 64 \times 1 \times 1$, stride: $(1,1)$
conv2: $64 \times 64 \times 3 \times 3$, stride: $(1,1)$; padding: $(1,1)$
conv3: $64 \times 256 \times 1 \times 1$, stride: $(1,1)$ | $\times 6$ |
| md8 | avgpool
fc: $256 \times 10$ | $\times 1$
$\times 1$ |

Table 10: Architectural details of tiers in the experiment with 7 tiers ($M = 7$). Client-side models include avgpool and f.c. layers as auxiliary components.

| Tier ($m$) | 1 | 2 | 3 | 4 | 5 | 6 | 7 |
|---|---|---|---|---|---|---|---|
| Client-side | md1 | md1 | md1 | md1 | md1 | md1 | md1 |
| | | md2 | md2 | md2 | md2 | md2 | md2 |
| | | | md3 | md3 | md3 | md3 | md3 |
| | | | | md4 | md4 | md4 | md4 |
| | | | | | md5 | md5 | md5 |
| | | | | | | md6 | md6 |
| | | | | | | | md7 |
| | avgpool | avgpool | avgpool | avgpool | avgpool | avgpool | avgpool |
| (f.c.) | $16 \times 10$ | $64 \times 10$ | $64 \times 10$ | $128 \times 10$ | $128 \times 10$ | $256 \times 10$ | $256 \times 10$ |
| Server-side | md2 | | | | | | |
| | md3 | md3 | | | | | |
| | md4 | md4 | md4 | | | | |
| | md5 | md5 | md5 | md5 | | | |
| | md6 | md6 | md6 | md6 | md6 | | |
| | md7 | md7 | md7 | md7 | md7 | md7 | |
| | md8 | md8 | md8 | md8 | md8 | md8 | md8 |

Table 11: Modules in each tier for experiments with varying numbers of tiers ($M$).

| # Tiers ($M$) | Tier | Client-side | Server-side |
|---|---|---|---|
| 1 | 1 | md1, md2, md3, md4, md5, md6, md7 | md8 |
| 2 | 1 | md1, md2, md3, md4, md5, md6 | md7, md8 |
| | 2 | md1, md2, md3, md4, md5, md6, md7 | md8 |
| 3 | 1 | md1, md2,md3, md4, md5 | md6, md7, md8 |
| | 2 | md1, md2 , md3, md4, md5, md6 | md7, md8 |
| | 3 | md1, md2 , md3, md4, md5, md6, md7 | md8 |
| 4 | 1 | md1, md2, md3, md4 | md5, md6, md7, md8 |
| | 2 | md1, md2,md3, md4, md5 | md6, md7, md8 |
| | 3 | md1, md2 , md3, md4, md5, md6 | md7, md8 |
| | 4 | md1, md2 , md3, md4, md5, md6, md7 | md8 |
| 5 | 1 | md1, md2,md3 | md4, md5, md6, md7, md8 |
| | 2 | md1, md2, md3, md4 | md5, md6, md7, md8 |
| | 3 | md1, md2,md3, md4, md5 | md6, md7, md8 |
| | 4 | md1, md2 , md3, md4, md5, md6 | md7, md8 |
| | 5 | md1, md2 , md3, md4, md5, md6, md7 | md8 |
| 6 | 1 | md1, md2 | md3, md4, md5, md6, md7, md8 |
| | 2 | md1, md2,md3 | md4, md5, md6, md7, md8 |
| | 3 | md1, md2, md3, md4 | md5, md6, md7, md8 |
| | 4 | md1, md2,md3, md4, md5 | md6, md7, md8 |
| | 5 | md1, md2 , md3, md4, md5, md6 | md7, md8 |
| | 6 | md1, md2 , md3, md4, md5, md6, md7 | md8 |
| 7 | 1 | md1 | md2, md3, md4, md5, md6, md7, md8 |
| | 2 | md1, md2 | md3, md4, md5, md6, md7, md8 |
| | 3 | md1, md2,md3 | md4, md5, md6, md7, md8 |
| | 4 | md1, md2, md3, md4 | md5, md6, md7, md8 |
| | 5 | md1, md2,md3, md4, md5 | md6, md7, md8 |
| | 6 | md1, md2 , md3, md4, md5, md6 | md7, md8 |
| | 7 | md1, md2 , md3, md4, md5, md6, md7 | md8 |

## B    PROOF OF THEOREM 1

### B.1    ADDITIONAL DEFINITIONS

We provide precise definitions of assumptions as follows:

**Assumption 1 (A1: $L$-smoothness)** *The loss function is differentiable and $L$-smooth, i.e.,* $\|\nabla f(\boldsymbol{u}) - \nabla f(\mathbf{v})\| \le L\|\boldsymbol{u} - \mathbf{v}\|, \forall f, \boldsymbol{u}, \mathbf{v}.$

**Assumption 2 (A2: Bounded gradients)** *The expected squared norm of stochastic gradients of each objective function is upper bounded:* $\mathbb{E}\|\nabla f(\boldsymbol{u})\|^2 \le G_1^2, \forall f, \boldsymbol{u}.$

**Assumption 3 (A3: Bounded variance)** *The stochastic gradient* $\mathbf{g}(\mathbf{x}) := \nabla f(\boldsymbol{u})$ *is unbiased,* $\mathbb{E}[\mathbf{g}_i(\boldsymbol{u})] = \nabla f_i(\boldsymbol{u})$, *and has bounded variance* $\mathbb{E}[\|\mathbf{g}_i(\boldsymbol{u}) - \nabla f_i(\boldsymbol{u})|^2] \le \sigma^2, \forall f, \boldsymbol{u}.$

**Assumption 4 (A4: $\mu$-convex)** *$f$ is $\mu$-convex for $\mu \ge 0$, and it satisfies:* $f(\boldsymbol{u}) + \nabla^T f(\boldsymbol{u})(\boldsymbol{v} - \boldsymbol{u}) + \frac{\mu}{2}\|\boldsymbol{v} - \boldsymbol{u}\|^2 \le f(\boldsymbol{v}), \forall f, \boldsymbol{u}, \boldsymbol{v}.$

**Assumption 5 (A5: Bounded gradient dissimilarity)** *For both client and server sides and all tier models, there are constants $G_2 \ge 0$; $B \ge 1$ such that $\frac{1}{K}\sum_{i=1}^{K} \|\nabla f_i(\boldsymbol{u})\|^2 \le G_2^2 + B^2\|\nabla f(\boldsymbol{u})\|^2, \forall \boldsymbol{u}.$*

*If $\{f_i\}$ are convex, we can relax the assumption to* $\frac{1}{K}\sum_{i=1}^{K}\|\nabla f_i(\boldsymbol{u})\|^2 \le G_2^2 + 2LB^2(f(\boldsymbol{u}) - f^\star), \forall \boldsymbol{u}.$

**Assumption 6 (A6: Bounded distance)** *The time-varying parameter satisfies $d_m^{c_m^{(r)}} < \infty \ \forall m, r.$*

### B.2    KEY LEMMAS

To make our proof clear, we introduce some useful lemmas.

**Lemma 1 (linear convergence rate, lemma 1 of Karimireddy et al. (2020))** *For every non-negative sequence $\{d_{r-1}\}_{r\ge 1}$ and any parameters $\mu > 0, \eta_{\max} \in (0, 1/\mu], q \ge 0, R \ge \frac{1}{2\eta_{\max}\mu}$, there exists a constant step-size $\eta \le \eta_{\max}$ and weights $w_r := (1 - \mu\eta)^{1-r}$ such that for $W_R := \sum_{r=1}^{R+1} w_r$*
$$\Psi_R := \frac{1}{W_R}\sum_{r=1}^{R+1}\left(\frac{w_r}{\eta}(1-\mu\eta)d_{r-1} - \frac{w_r}{\eta}d_r + q\eta w_r\right) = \mathcal{O}\left(\mu d_0 \exp\left(-\mu\eta_{\max}R\right) + \frac{q}{\mu R}\right).$$

**Lemma 2 (convergence rate on non-convex functions, lemma 2 of Karimireddy et al. (2020))** *For every non-negative sequence $\{d_{r-1}\}_{r\ge 1}$ and any parameters $\eta_{\max} \ge 0, q \ge 0, R \ge 0$, there exists a constant step-size $\eta \le \eta_{\max}$ and weights $w_r = 1$ such that,*

$$\Psi_R := \frac{1}{R+1}\sum_{r=1}^{R+1}\left(\frac{d_{r-1}}{\eta} - \frac{d_r}{\eta} + q_1\eta + q_2\eta^2\right) \le$$

$$\frac{d_0}{\eta_{\max}(R+1)} + \frac{2\sqrt{q_1 d_0}}{\sqrt{R+1}} + 2\left(\frac{d_0}{R+1}\right)^{\frac{2}{3}}q_2^{\frac{1}{3}}.$$

**Lemma 3 (Relaxed triangle inequality, lemma 3 of Karimireddy et al. (2020))** *Let $\{v_1, \ldots, v_\tau\}$ be $\tau$ vectors in $\mathbb{R}^d$. Then the following are true:*

   1. $\|v_i + v_j\|^2 \le (1 + a)\|v_i\|^2 + \left(1 + \frac{1}{a}\right)\|v_j\|^2$ *for any $a > 0$, and*

   2. $\|\sum_{i=1}^{\tau} v_i\|^2 \le \tau \sum_{i=1}^{\tau}\|v_i\|^2.$

**Lemma 4 (separating mean and variance, lemma 4 of Karimireddy et al. (2020))** *Let $\{\Xi_1, \ldots, \Xi_\kappa\}$ be $\kappa$ random variables in $\mathbb{R}^d$ which are not necessarily independent. First*

*suppose that their mean is $\mathbb{E}[\Xi_i] = \xi_i$ and variance is bounded as $\mathbb{E}\left[\|\Xi_i - \xi_i\|^2\right] \leq \sigma^2$. Then, the following holds*

$$\mathbb{E}\left[\left\|\sum_{i=1}^{\kappa} \Xi_i\right\|^2\right] \leq \left\|\sum_{i=1}^{\kappa} \xi_i\right\|^2 + \kappa^2 \sigma^2$$

*Now, let's consider the conditional mean of the variables $\Xi_i$ is denoted as $E[\Xi_i|\Xi_{i-1}, \ldots, \Xi_1] = \xi_i$. Then,*

$$\mathbb{E}\left[\left\|\sum_{i=1}^{\kappa} \Xi_i\right\|^2\right] \leq 2\left\|\sum_{i=1}^{\kappa} \xi_i\right\|^2 + 2\kappa\sigma^2$$

**Lemma 5 (perturbed strong convexity, lemma 5 of Karimireddy et al. (2020))** *The following holds for any $L$-smooth and $\mu$-strongly convex function $h$, and any $\boldsymbol{u}, \boldsymbol{v}, \boldsymbol{w}$ in the domain of $h$:*

$$\langle \nabla h(\boldsymbol{u}), \boldsymbol{w} - \boldsymbol{v} \rangle \geq h(\boldsymbol{w}) - h(\boldsymbol{v}) + \frac{L}{4}\|\boldsymbol{v} - \boldsymbol{w}\|^2 - L\|\boldsymbol{w} - \boldsymbol{u}\|^2$$

### B.3 PROOF OF CONVERGENCE

We prove the rate of convergence for both client-side and server-side convex functions in the following. The proof for non-convex functions follows similar steps and is easy to derive using the techniques in the rest of the paper.

#### B.3.1 CONVEX FUNCTIONS

**Client-side model convergence.** Initially, we demonstrate client-side function convergence. Suppose that client-side functions satisfy the following assumptions: (1), (3), (4), and (5). The update of the model satisfies the following:

$$\Delta \boldsymbol{w}^{c_m} = -\frac{\eta}{A^{c_m^{(r)}}} \sum_{k \in \mathcal{A}^{c_m^{(r)}}} g_k^{c_m}(\boldsymbol{w}_k^{c_m}) \Rightarrow \mathbb{E}[\Delta \boldsymbol{w}^{c_m}] = -\frac{\eta}{K} \sum_k \mathbb{E}\left[\nabla f_k^{c_m}(\boldsymbol{w}_k^{c_m})\right].$$

where $g_k^{c_m}(\cdot)$ is unbiased stochastic gradient of $\nabla f_k^{c_m}(\cdot)$. We implicitly incorporate auxiliary layers $\boldsymbol{w}^{a_m}$ in $\boldsymbol{w}_k^{c_m}$ for simplicity in the proof. We define $A^{c_m^{(r)}}$ and $\mathcal{A}^{c_m^{(r)}}$ as the number and the set of clients in tier $m$ at round $r$. We denote the expectation over all the randomness generated in the prior round, $r$, using $\mathbb{E}$. According to the above observation, we proceed as follows:

$$\mathbb{E}\left\|\boldsymbol{w}^{c_m} + \Delta \boldsymbol{w}^{c_m} - \boldsymbol{w}^{c_m^\star}\right\|^2 = \left\|\boldsymbol{w}^{c_m} - \boldsymbol{w}^{c_m^\star}\right\|^2 - \frac{2\eta}{K}\sum_k \left\langle \nabla f_k^{c_m}(\boldsymbol{w}_k^{c_m}), \boldsymbol{w}^{c_m} - \boldsymbol{w}^{c_m^\star} \right\rangle$$

$$+ \eta^2 \mathbb{E}\left\|\frac{1}{A^{c_m^{(r)}}} \sum_{k \in \mathcal{A}^{c_m^{(r)}}} g_k^{c_m}(\boldsymbol{w}_k^{c_m})\right\|^2$$

$$\overset{\text{Lem. 4}}{\leq} \left\|\boldsymbol{w}^{c_m} - \boldsymbol{w}^{c_m^\star}\right\|^2 \underbrace{- \frac{2\eta}{K}\sum_k \left\langle \nabla f_k^{c_m}(\boldsymbol{w}_k^{c_m}), \boldsymbol{w}^{c_m} - \boldsymbol{w}^{c_m^\star} \right\rangle}_{\mathcal{T}_1}$$

$$+ \underbrace{\eta^2 \mathbb{E}\left\|\frac{1}{A^{c_m^{(r)}}} \sum_{k \in \mathcal{A}^{c_m^{(r)}}} \nabla f_k^{c_m}(\boldsymbol{w}_k^{c_m})\right\|^2}_{\mathcal{T}_2} + \frac{\eta^2 \sigma^2}{A^{c_m^{(r)}}}.$$

$$(7)$$

By using Lemma 5 with $h = f_k^{c_m}$, $\boldsymbol{u} = \boldsymbol{w}_k^{c_m}$, $\boldsymbol{v} = \boldsymbol{w}^{c_m^\star}$, and $\boldsymbol{w} = \boldsymbol{w}^{c_m}$ to the first term $\mathcal{T}_1$.

$$
\begin{aligned}
\mathcal{T}_1 &= \frac{2\eta}{K} \sum_k \left\langle \nabla f_k^{c_m}(\boldsymbol{w}_k^{c_m}), \boldsymbol{w}^{c_m^\star} - \boldsymbol{w}^{c_m} \right\rangle \\
&\leq \frac{2\eta}{K} \sum_k \left( f_k^{c_m}\left(\boldsymbol{w}^{c_m^\star}\right) - f_k^{c_m}(\boldsymbol{w}^{c_m}) + L \left\| \boldsymbol{w}_k^{c_m} - \boldsymbol{w}^{c_m} \right\|^2 - \frac{\mu}{4} \left\| \boldsymbol{w}^{c_m} - \boldsymbol{w}^{c_m^\star} \right\|^2 \right) \\
&= -2\eta \left( f^{c_m}(\boldsymbol{w}^{c_m}) - f^{c_m}\left(\boldsymbol{w}^{c_m^\star}\right) + \frac{\mu}{4} \left\| \boldsymbol{w}^{c_m} - \boldsymbol{w}^{c_m^\star} \right\|^2 \right)
\end{aligned}
$$

While DTFL can be used with more than one local epoch, to simplify the discussion, we will consider the case where the local epoch is equal to 1. In this case, the last equality holds because $\boldsymbol{w}_k^{c_m} = \boldsymbol{w}^{c_m}$.

To evaluate $\mathcal{T}_2$, we utilize the relaxed triangle inequality repeatedly (Lemma 3).

$$
\begin{aligned}
\mathcal{T}_2 &= \eta^2 \mathbb{E} \left\| \frac{1}{A^{c_m^{(r)}}} \sum_{k \in \mathcal{A}^{c_m^{(r)}}} \left[ \nabla f_k^{c_m}(\boldsymbol{w}_k^{c_m}) - \nabla f_k^{c_m}(\boldsymbol{w}^{c_m}) + \nabla f_k^{c_m}(\boldsymbol{w}^{c_m}) \right] \right\|^2 \\
&\leq 2\eta^2 \mathbb{E} \left\| \frac{1}{A^{c_m^{(r)}}} \sum_{k \in \mathcal{A}^{c_m^{(r)}}} \left[ \nabla f_k^{c_m}(\boldsymbol{w}_k^{c_m}) - \nabla f_k^{c_m}(\boldsymbol{w}^{c_m}) \right] \right\|^2 \\
&\quad + 2\eta^2 \mathbb{E} \left\| \frac{1}{A^{c_m^{(r)}}} \sum_{k \in \mathcal{A}^{c_m^{(r)}}} \nabla f_k^{c_m}(\boldsymbol{w}^{c_m}) \right\|^2 \\
&\leq \frac{2\eta^2}{K} \sum_k \mathbb{E} \left\| \nabla f_k^{c_m}(\boldsymbol{w}_k^{c_m}) - \nabla f_k^{c_m}(\boldsymbol{w}^{c_m}) \right\|^2 \\
&\quad + 2\eta^2 \mathbb{E} \left\| \frac{1}{A^{c_m^{(r)}}} \sum_{k \in \mathcal{A}^{c_m^{(r)}}} \nabla f_k^{c_m}(\boldsymbol{w}^{c_m}) - \nabla f^{c_m}(\boldsymbol{w}^{c_m}) + \nabla f^{c_m}(\boldsymbol{w}^{c_m}) \right\|^2 \\
&\leq \frac{2\eta^2 L^2}{K} \sum_k \mathbb{E} \left\| \boldsymbol{w}_k^{c_m} - \boldsymbol{w}^{c_m} \right\|^2 \\
&\quad + 2\eta^2 \| \nabla f(\boldsymbol{w}^{c_m}) \|^2 + \left( 1 - \frac{A^{c_m^{(r)}}}{K} \right) 4\eta^2 \frac{1}{A^{c_m^{(r)}} K} \sum_k \| \nabla f_k^{c_m}(\boldsymbol{w}^{c_m}) \|^2 \\
&\overset{Assump.5}{\leq} \frac{2\eta^2 L^2}{K} \sum_k \mathbb{E} \left\| \boldsymbol{w}_k^{c_m} - \boldsymbol{w}^{c_m} \right\|^2 + 8\eta^2 L \left( B^2 + 1 \right) \left( f^{c_m}(\boldsymbol{w}^{c_m}) - f^{c_m}\left( \boldsymbol{w}^{c_m^\star} \right) \right) \\
&\quad + \left( 1 - \frac{A^{c_m^{(r)}}}{K} \right) \frac{4\eta^2}{A^{c_m^{(r)}}} G_2^2 \\
&= 8\eta^2 L \left( B^2 + 1 \right) \left( f^{c_m}(\boldsymbol{w}^{c_m}) - f^{c_m}\left( \boldsymbol{w}^{c_m^\star} \right) \right) + \left( 1 - \frac{A^{c_m^{(r)}}}{K} \right) \frac{4\eta^2}{A^{c_m^{(r)}}} G_2^2
\end{aligned}
$$

Substituting the obtained bounds for $\mathcal{T}_1$ and $\mathcal{T}_2$ back into the equation (7),

$$\mathbb{E}\left\|\boldsymbol{w}^{c_m} + \Delta\boldsymbol{w}^{c_m} - \boldsymbol{w}^{c_m^\star}\right\|^2 \leq \left(1 - \frac{\mu\eta}{2}\right)\left\|\boldsymbol{w}^{c_m} - \boldsymbol{w}^{c_m^\star}\right\|^2$$
$$- \left(2\eta - 8L\eta^2\left(B^2+1\right)\right)\left(f^{c_m}(\boldsymbol{w}^{c_m}) - f^{c_m}(\boldsymbol{w}^{c_m^\star})\right)$$
$$+ \frac{1}{A^{c_m^{(r)}}}\eta^2\sigma^2 + \left(1 - \frac{A^{c_m^{(r)}}}{K}\right)\frac{4\eta^2}{A^{c_m^{(r)}}}G_2^2$$

Moving the $(f^{c_m}(\boldsymbol{w}^{c_m}) - f^{c_m}(\boldsymbol{w}^{c_m^\star}))$ term,

$$\left(2\eta - 8L\eta^2\left(B^2+1\right)\right)\left(f^{c_m}(\boldsymbol{w}^{c_m}) - f^{c_m}(\boldsymbol{w}^{c_m^\star})\right) \leq \left(1 - \frac{\mu\eta}{2}\right)\left\|\boldsymbol{w}^{c_m} - \boldsymbol{w}^{c_m^\star}\right\|^2$$
$$- \mathbb{E}\left\|\boldsymbol{w}^{c_m} + \Delta\boldsymbol{w}^{c_m} - \boldsymbol{w}^{c_m^\star}\right\|^2$$
$$+ \frac{1}{A^{c_m^{(r)}}}\eta^2\sigma^2 + \left(1 - \frac{A^{c_m^{(r)}}}{K}\right)\frac{4\eta^2}{A^{c_m^{(r)}}}G_2^2$$
$$= \left(1 - \frac{\mu\eta}{2}\right)\left\|\boldsymbol{w}^{c_m^{(r)}} - \boldsymbol{w}^{c_m^\star}\right\|^2$$
$$- \left\|\boldsymbol{w}^{c_m^{(r+1)}} - \boldsymbol{w}^{c_m^\star}\right\|^2$$
$$+ \frac{1}{A^{c_m^{(r)}}}\eta^2\sigma^2 + \left(1 - \frac{A^{c_m^{(r)}}}{K}\right)\frac{4\eta^2}{A^{c_m^{(r)}}}G_2^2$$

Considering $8L\eta\left(B^2+1\right) \leq 1$, and divide by $\eta$ yields,

$$f^{c_m}(\boldsymbol{w}^{c_m}) - f^{c_m}(\boldsymbol{w}^{c_m^\star}) \leq \frac{1}{\eta}\left(1 - \frac{\mu\eta}{2}\right)\left\|\boldsymbol{w}^{c_m^{(r)}} - \boldsymbol{w}^{c_m^\star}\right\|^2 - \frac{1}{\eta}\left\|\boldsymbol{w}^{c_m^{(r+1)}} - \boldsymbol{w}^{c_m^\star}\right\|^2$$
$$+ \eta\left[\frac{1}{A^{c_m^{(r)}}}\sigma^2 + \left(1 - \frac{A^{c_m^{(r)}}}{K}\right)\frac{4}{A^{c_m^{(r)}}}G_2^2\right]$$

By applying Lemma 1 with $q = \left(\frac{\sigma^2}{A^{c_m^{(r)}}} + \left(1 - \frac{A^{c_m^{(r)}}}{K}\right)\frac{4G_2^2}{A^{c_m^{(r)}}}\right)$, which holds true for $R \geq \frac{1}{2\eta_{\max}\mu}$, and considering $8L\eta\left(B^2+1\right) \leq 1$, we can rewrite the bound for $R$ as $R \geq \frac{4L(1+B^2)}{\mu}$. Therefore, we obtain,

$$\mathbb{E}\left[f^{c_m}(\overline{\boldsymbol{w}^{c_m}}^R)\right] - f^{c_m}(\boldsymbol{w}^{c_m^\star}) \leq \left\|\boldsymbol{w}^{c_m^0} - \boldsymbol{w}^{c_m^\star}\right\|^2\mu\exp\left(-\frac{\eta}{2}\mu R\right)$$
$$+ \frac{\eta}{\mu R}\left(\frac{\sigma^2}{A^{c_m^{(r)}}} + \left(1 - \frac{A^{c_m^{(r)}}}{K}\right)\frac{4G_2^2}{A^{c_m^{(r)}}}\right)$$

We derive the desired learning rate for the client-side objective function of tier $m$, which relies on $A^{c_m^{(r)}}$. Notably, a tier with a larger number of clients experiences faster convergence. Therefore, when more clients are assigned to a tier, it converges in fewer rounds. However, the total training time depends on various factors discussed in Section 3. To establish an upper bound on the convergence rate for each tier, we introduce the notation $A^m = \min_r\{A^{c_m^{(r)}} > 0\}$. Consequently, we can determine the following convergence rates:

$$\mathbb{E}\left[f^{c_m}(\overline{\boldsymbol{w}^{c_m}}^R)\right] - f^{c_m}(\boldsymbol{w}^{c_m^\star}) \leq \underbrace{\left\|\boldsymbol{w}^{c_m^0} - \boldsymbol{w}^{c_m^\star}\right\|^2}_{:=D^2} \mu \exp\left(-\frac{\eta}{2}\mu R\right)$$
$$+ \frac{\eta}{\mu R}\left(\frac{\sigma^2}{A^m} + \left(1 - \frac{A^m}{K}\right)\frac{4G_2^2}{A^m}\right)$$

Utilizing asymptotic notation, we obtain the following expression for the client-side convergence rate:

$$\mathbb{E}\left[f^{c_m}(\overline{\boldsymbol{w}^{c_m}}^R)\right] - f^{c_m}(\boldsymbol{w}^{c\star}) = \mathcal{O}\left(\mu D^2 \exp\left(-\frac{\eta}{2}\mu R\right) + \frac{\eta H_1^2}{\mu R A^m}\right)$$

where $H_1^2 := \sigma^2 + \left(1 - \frac{A^m}{K}\right)G_2^2$, and $D := \left\|\boldsymbol{w}^{c_m^0} - \boldsymbol{w}^{c_m^\star}\right\|$. The global model converges once all tiers have converged, with the convergence rate being determined by the tier with the slowest rate of convergence.

**Server-side model convergence.** Now, we demonstrate the server-side non-convex convergence rate and the corresponding rate for the convex function can be derived using the previously described technique and by applying Lemma 1.

Suppose that server-side functions satisfy the following Assumptions: 1, 2, 3, 5, and 6. The update of the model satisfies the following:

$$\Delta\boldsymbol{w}^{s_m} = -\frac{\eta}{A^{c_m^{(r)}}}\sum_{k \in \mathcal{A}^{c_m^{(r)}}} g_k^{s_m}(\boldsymbol{w}_k^{s_m}) \Rightarrow \mathbb{E}[\Delta\boldsymbol{w}^{s_m}] = -\frac{\eta}{K}\sum_k \mathbb{E}\left[\nabla f_k^{s_m}(\boldsymbol{z}_k^{c_m}; \boldsymbol{w}_k^{s_m})\right].$$

Based on L-smoothness Assumption 1, we have:

$$f_k^{s_m}\left(\boldsymbol{z}_k^{c_m}; \boldsymbol{w}_k^{s_m^{(r+1)}}\right) \leq f_k^{s_m}\left(\boldsymbol{z}_k^{c_m}; \boldsymbol{w}_k^{s_m^{(r)}}\right) + \nabla f_k^{s_m}\left(\boldsymbol{z}_k^{c_m}; \boldsymbol{w}_k^{s_m^{(r)}}\right)^T (\boldsymbol{w}_k^{s_m^{(r+1)}} - \boldsymbol{w}_k^{s_m^{(r)}}) + \frac{L}{2}\|\boldsymbol{w}_k^{s_m^{(r+1)}} - \boldsymbol{w}_k^{s_m^{(r)}}\|.$$

We utilize the weight update formula and incorporate it into the above inequality. Then, by taking the expectation across all randomness, we arrive at the following.

$$\mathbb{E}\left[f^{s_m}(\boldsymbol{z}^{c_m^{(r)}}; \boldsymbol{w}^{s_m^{(r+1)}})\right] \leq \mathbb{E}\left[f^{s_m}(\boldsymbol{z}^{c_m^{(r)}}; \boldsymbol{w}^{s_m^{(r)}})\right] - \eta\underbrace{\mathbb{E}\left[\nabla f^{s_m}(\boldsymbol{z}^{c_m^{(r)}}; \boldsymbol{w}^{s_m^{(r)}})^T\left(\frac{1}{A^{c_m^{(r)}}}\sum_{k \in \mathcal{A}^{c_m^{(r)}}}\nabla f_k^{s_m}\left(\boldsymbol{z}^{c_m^{(r)}}; \boldsymbol{w}^{s_m^{(r)}}\right)\right)\right]}_{\mathcal{T}_3}$$

$$+ \frac{L}{2}\eta^2\underbrace{\mathbb{E}\left\|\frac{1}{A^{c_m^{(r)}}}\sum_{k \in \mathcal{A}^{c_m^{(r)}}}\nabla f_k^{s_m}\left(\boldsymbol{z}_k^{c_m}; \boldsymbol{w}_k^{s_m}\right)\right\|^2}_{\mathcal{T}_4}.$$

$$(8)$$

We now demonstrate that both $\mathcal{T}_3$ and $\mathcal{T}_4$ are bounded. Using Assumption 4 and adopting a similar approach to the one used for the client-side function, we can establish that $\mathcal{T}_4$ is bounded:

$$\eta^2\mathbb{E}\left[\left\|\frac{1}{A^{c_m^{(r)}}}\sum_{k \in \mathcal{A}^{c_m^{(r)}}}\nabla f_k^{s_m}\left(\boldsymbol{z}_k^{c_m}; \boldsymbol{w}_k^{s_m}\right)\right\|^2\right] \leq$$

$$8\eta^2 L\left(B^2 + 1\right)\left(f_k^{s_m}(\boldsymbol{w}_k^{s_m}) - f_k^{s_m}\left(\boldsymbol{w}_k^{s_m^\star}\right)\right) + \left(1 - \frac{A^{c_m^{(r)}}}{K}\right)\frac{4\eta^2}{A^{c_m^{(r)}}}G_2^2$$

To show $\mathcal{T}_3$ is bounded, we consider the following inequality:

$$
\left\| \frac{1}{A^{c_m^{(r)}}} \sum_{k \in \mathcal{A}^{c_m^{(r)}}} \left\| \nabla f^{s_m}(\boldsymbol{w}^{s_m^{(r)}}) \right\|^2 - \underbrace{\mathbb{E}\left[ \nabla f^{s_m}(\boldsymbol{w}^{s_m^{(r)}})^T \left( \frac{1}{A^{c_m^{(r)}}} \sum_{k \in \mathcal{A}^{c_m^{(r)}}} \nabla f_k^{s_m}\left(\boldsymbol{z}^{c_m^{(r)}}; \boldsymbol{w}^{s_m^{(r)}}\right) \right) \right]}_{\mathcal{T}_3} \right\|
$$

$$
= \left\| \frac{1}{A^{c_m^{(r)}}} \sum_{k \in \mathcal{A}^{c_m^{(r)}}} \left\| \nabla f^{s_m}(\boldsymbol{w}^{s_m^{(r)}}) \right\|^2 - \frac{1}{A^{c_m^{(r)}}} \sum_{k \in \mathcal{A}^{c_m^{(r)}}} \nabla f^{s_m}(\boldsymbol{w}^{s_m^{(r)}})^T \mathbb{E}\left[ \nabla f_k^{s_m}\left(\boldsymbol{z}^{c_m^{(r)}}; \boldsymbol{w}^{s_m^{(r)}}\right) \right] \right\|
$$

$$
= \left[ \left\| \nabla f^{s_m}(\boldsymbol{w}^{s_m^{(r)}})^T \frac{1}{A^{c_m^{(r)}}} \sum_{k \in \mathcal{A}^{c_m^{(r)}}} \nabla f^{s_m}(\boldsymbol{w}^{s_m^{(r)}}) - \frac{1}{A^{c_m^{(r)}}} \sum_{k \in \mathcal{A}^{c_m^{(r)}}} \nabla f^{s_m}(\boldsymbol{w}^{s_m^{(r)}})^T \mathbb{E}\left[ \nabla f_k^{s_m}\left(\boldsymbol{z}^{c_m^{(r)}}; \boldsymbol{w}^{s_m^{(r)}}\right) \right] \right\| \right]
$$

$$
= \left[ \left\| \nabla f^{s_m}(\boldsymbol{w}^{s_m^{(r)}})^T \left( \frac{1}{A^{c_m^{(r)}}} \sum_{k \in \mathcal{A}^{c_m^{(r)}}} \nabla f^{s_m}(\boldsymbol{w}^{s_m^{(r)}}) - \frac{1}{A^{c_m^{(r)}}} \sum_{k \in \mathcal{A}^{c_m^{(r)}}} \mathbb{E}\left[ \nabla f_k^{s_m}\left(\boldsymbol{z}^{c_m^{(r)}}; \boldsymbol{w}^{s_m^{(r)}}\right) \right] \right) \right\| \right]
$$

$$
\overset{(a)}{\leq} \left[ \left\| \nabla f^{s_m}(\boldsymbol{w}^{s_m^{(r)}})^T \right\| \left\| \frac{1}{A^{c_m^{(r)}}} \sum_{k \in \mathcal{A}^{c_m^{(r)}}} \left( \nabla f^{s_m}(\boldsymbol{w}^{s_m^{(r)}}) - \mathbb{E}\left[ \nabla f_k^{s_m}\left(\boldsymbol{z}^{c_m^{(r)}}; \boldsymbol{w}^{s_m^{(r)}}\right) \right] \right) \right\| \right]
$$

$$
\leq \underbrace{\sqrt{\left[ \left\| \nabla f^{s_m}(\boldsymbol{w}^{s_m^{(r)}}) \right\|^2 \right]}}_{\mathcal{T}_5} \underbrace{\left\| \frac{1}{A^{c_m^{(r)}}} \sum_{k \in \mathcal{A}^{c_m^{(r)}}} \left( \nabla f^{s_m}(\boldsymbol{w}^{s_m^{(r)}}) - \mathbb{E}\left[ \nabla f_k^{s_m}\left(\boldsymbol{z}^{c_m^{(r)}}; \boldsymbol{w}^{s_m^{(r)}}\right) \right] \right) \right\|}_{\mathcal{T}_6}
$$

$$(9)$$

where we used Cauchy-Schwartz inequality in step (a). To show $\mathcal{T}_3$ is bounded, we begin to show $\mathcal{T}_5$ and $\mathcal{T}_6$ are bounded. Based on Assumption 2 the $\mathcal{T}_5$ is bounded as $\mathcal{T}_5 \leq \sqrt{G_1^2} = G_1$.

To show $\mathcal{T}_6$ is bounded, we have:

$$
\left\| \frac{1}{A^{c_m^{(r)}}} \sum_{k \in \mathcal{A}^{c_m^{(r)}}} \left( \nabla f^{s_m}(\boldsymbol{w}^{s_m^{(r)}}) - \mathbb{E}\left[ \nabla f_k^{s_m}\left(\boldsymbol{z}^{c_m^{(r)}}; \boldsymbol{w}^{s_m^{(r)}}\right) \right] \right) \right\|
$$

$$
= \left\| \frac{1}{A^{c_m^{(r)}}} \sum_{k \in \mathcal{A}^{c_m^{(r)}}} \left( \mathbb{E}\left[ \nabla f_k^{s_m}\left(\boldsymbol{z}^{c_m^{(r)}}; \boldsymbol{w}^{s_m^{(r)}}\right) - \nabla f^{s_m}(\boldsymbol{w}^{s_m^{(r)}}) \right] \right) \right\|
$$

$$
\leq \frac{1}{A^{c_m^{(r)}}} \sum_{k \in \mathcal{A}^{c_m^{(r)}}} \left\| \mathbb{E}\left[ \nabla f_k^{s_m}\left(\boldsymbol{z}^{c_m^{(r)}}; \boldsymbol{w}^{s_m^{(r)}}\right) - \nabla f^{s_m}(\boldsymbol{w}^{s_m^{(r)}}) \right] \right\|
$$

$$(10)$$

$$
= \frac{1}{A^{c_m^{(r)}}} \sum_{k \in \mathcal{A}^{c_m^{(r)}}} \left\| \int \nabla f_k^{s_m}\left(\boldsymbol{z}; \boldsymbol{w}^{s_m^{(r)}}\right) p^{c_m^{(r)}}(\boldsymbol{z}) d\boldsymbol{z} - \int \nabla f_k^{s_m}\left(\boldsymbol{z}; \boldsymbol{w}^{s_m^{(r)}}\right) p^{c_m^{(\star)}}(\boldsymbol{z}) d\boldsymbol{z} \right\|
$$

$$
\leq \frac{1}{K} \sum_k \int \left\| \nabla f_k^{s_m}\left(\boldsymbol{z}; \boldsymbol{w}^{s_m^{(r)}}\right) \right\| \left| p^{c_m^{(r)}}(\boldsymbol{z}) - p^{c_m^{(\star)}}(\boldsymbol{z}) \right| d\boldsymbol{z}
$$

$$
= \frac{1}{K} \sum_k \int \left( \left\| \nabla f_k^{s_m}\left(\boldsymbol{z}; \boldsymbol{w}^{s_m^{(r)}}\right) \right\| \sqrt{\left| p^{c_m^{(r)}}(\boldsymbol{z}) - p^{c_m^{(\star)}}(\boldsymbol{z}) \right|} \right) \sqrt{\left| p^{c_m^{(r)}}(\boldsymbol{z}) - p^{c_m^{(\star)}}(\boldsymbol{z}) \right|} d\boldsymbol{z}
$$

where we define the output of the client-side model $\boldsymbol{z}^{c_m^{(r)}}$ which follows the density function $p^{c_m^{(r)}}(\boldsymbol{z})$, with the converged density of the client-side represented as $p^{c_m^{(\star)}}(\boldsymbol{z})$. By applying the Cauchy-Swchartz inequality again, we observe that:

$$
\left\| \frac{1}{A^{c_m^{(r)}}} \sum_{k \in \mathcal{A}^{c_m^{(r)}}} \left( \nabla f^{s_m}(\boldsymbol{w}^{s_m^{(r)}}) - \mathbb{E}\left[ \nabla f_k^{s_m}\left( \boldsymbol{z}^{c_m^{(r)}}; \boldsymbol{w}^{s_m^{(r)}} \right) \right) \right] \right\|
$$
$$
\leq \frac{1}{K} \sum_k \sqrt{\int \left( \left\| \nabla f_k^{s_m}\left( \boldsymbol{z}; \boldsymbol{w}^{s_m^{(r)}} \right) \right\|^2 \left| p^{c_m^{(r)}}(\boldsymbol{z}) - p^{c_m^{(\star)}}(\boldsymbol{z}) \right| \right) d\boldsymbol{z}} \sqrt{\left| p^{c_m^{(r)}}(\boldsymbol{z}) - p^{c_m^{(\star)}}(\boldsymbol{z}) \right| d\boldsymbol{z}}
$$
$$
= \frac{1}{K} \sum_k \sqrt{\int \left( \left\| \nabla f_k^{s_m}\left( \boldsymbol{z}; \boldsymbol{w}^{s_m^{(r)}} \right) \right\|^2 \left| p^{c_m^{(r)}}(\boldsymbol{z}) - p^{c_m^{(\star)}}(\boldsymbol{z}) \right| \right) d\boldsymbol{z}} \sqrt{d^{c_m^{(r)}}}
\tag{11}
$$

where $d^{c_m^{(r)}} \triangleq \int \left| p^{c_m^{(r)}}(\boldsymbol{z}) - p^{c_m^{(\star)}}(\boldsymbol{z}) \right| d\boldsymbol{z}$. To bound the inequality above, we proceed as follows:

$$
\int \left\| \nabla f_k^{s_m}\left( \boldsymbol{z}; \boldsymbol{w}^{s_m^{(r)}} \right) \right\|^2 \left| p^{c_m^{(r)}}(\boldsymbol{z}) - p^{c_m^{(\star)}}(\boldsymbol{z}) \right| d\boldsymbol{z}
$$
$$
\leq \int \left\| \nabla f_k^{s_m}\left( \boldsymbol{z}; \boldsymbol{w}^{s_m^{(r)}} \right) \right\|^2 \left( p^{c_m^{(r)}}(\boldsymbol{z}) + p^{c_m^{(\star)}}(\boldsymbol{z}) \right) d\boldsymbol{z}
\tag{12}
$$
$$
= \mathbb{E}\left[ \left\| \nabla f_k^{s_m}\left( \boldsymbol{z}^{c_m^{(r)}}; \boldsymbol{w}^{s_m^{(r)}} \right) \right\|^2 \right] + \mathbb{E}_{p^{c_m^{(\star)}}(\boldsymbol{z})}\left[ \left\| \nabla f_k^{s_m}\left( \boldsymbol{z}^{c_m^{(r)}}; \boldsymbol{w}^{s_m^{(r)}} \right) \right\|^2 \right]
$$
$$
\leq 2 \left( G_2^2 + 2LB^2(f^{s_m}(\boldsymbol{w}^{s_m^{(r)}}) - f^{s_m^\star}) \right)
$$

By substituting (11) and (12) into (10), we can observe:

$$
\mathcal{T}_6 = \left\| \mathbb{E}\left[ \frac{1}{A^{c_m^{(r)}}} \sum_{k \in \mathcal{A}^{c_m^{(r)}}} \left( \nabla f_k^{s_m}\left( \boldsymbol{z}^{c_m^{(r)}}; \boldsymbol{w}^{s_m^{(r)}} \right) - \nabla f^{s_m}(\boldsymbol{w}^{s_m^{(r)}}) \right) \right] \right\|
$$
$$
\leq \mathbb{E}\left[ \frac{1}{K} \sum_k \left\| \left( \nabla f_k^{s_m}\left( \boldsymbol{z}^{c_m^{(r)}}; \boldsymbol{w}^{s_m^{(r)}} \right) - \nabla f^{s_m}(\boldsymbol{w}^{s_m^{(r)}}) \right) \right\| \right]
$$
$$
\leq \frac{1}{K} \sum_k \sqrt{2 \left( G_2^2 + 2LB^2(f^{s_m}(\boldsymbol{w}^{s_m^{(r)}}) - f^{s_m^\star}) \right) d^{c_m^{(r)}}} \leq \sqrt{2 \left( G_2^2 + 2LB^2(f^{s_m}(\boldsymbol{w}^{s_m^{(r)}}) - f^{s_m^\star}) \right) d^{c_m^{(r)}}}
$$

Based on $\mathcal{T}_5$ and $\mathcal{T}_6$, we can write (9) as:

$$
\left| \frac{1}{A^{c_m^{(r)}}} \sum_{k \in \mathcal{A}^{c_m^{(r)}}} \left\| \nabla f^{s_m}(\boldsymbol{w}^{s_m^{(r)}}) \right\|^2 - \frac{1}{A^{c_m^{(r)}}} \sum_{k \in \mathcal{A}^{c_m^{(r)}}} \nabla f^{s_m}(\boldsymbol{w}^{s_m^{(r)}})^T \mathbb{E}\left[ \nabla f_k^{s_m}\left( \boldsymbol{z}^{c_m^{(r)}}; \boldsymbol{w}^{s_m^{(r)}} \right) \right] \right|
$$
$$
\leq \sqrt{2G_1^2(G_2^2 + 2LB^2(f^{s_m}(\boldsymbol{w}^{s_m^{(r)}}) - f^{s_m^\star}))d^{c_m^{(r)}}}
$$

Thus we obtain the following bound for $\mathcal{T}_3$:

$$
- \mathbb{E}\left[ \nabla f^{s_m}(\boldsymbol{w}^{s_m^{(r)}})^T \left( \frac{1}{A^{c_m^{(r)}}} \sum_{k \in \mathcal{A}^{c_m^{(r)}}} \nabla f_k^{s_m}\left( \boldsymbol{z}^{c_m^{(r)}}; \boldsymbol{w}^{s_m^{(r)}} \right) \right) \right]
$$
$$
\leq - \left( \mathbb{E}\left[ \left\| \nabla f^{s_m}(\boldsymbol{w}^{s_m^{(r)}}) \right\|^2 \right] - \sqrt{2G_1^2(G_2^2 + 2LB^2(f^{s_m}(\boldsymbol{w}^{s_m^{(r)}}) - f^{s_m^\star}))d^{c_m^{(r)}}} \right)
$$

As previously defined, $A^m = \min_r \{A^{c^{(r)}_m} > 0\}$. By employing the bounds from $\mathcal{T}_3$ and $\mathcal{T}_4$ in (8), we have:

$$
\mathbb{E}\left[f^{s_m}(\boldsymbol{w}^{s^{(r+1)}_m})\right] \leq \mathbb{E}\left[f^{s_m}(\boldsymbol{w}^{s^{(r)}_m})\right]
$$
$$
- \eta\left(\mathbb{E}\left[\left\|\nabla f^{s_m}(\boldsymbol{w}^{s^{(r)}_m})\right\|^2\right] - \sqrt{2G_1^2(G_2^2 + 2LB^2(f^{s_m}(\boldsymbol{w}^{s^{(r)}_m}) - f^{s^\star_m}))d^{c^{(r)}_m}}\right)
$$
$$
+ 2\eta^2 L^3\left(B^2 + 1\right)\left(f^{s_m}(\boldsymbol{w}^{s_m}) - f^{s^\star_m}\right) + \left(1 - \frac{A^m}{K}\right)\frac{\eta^2 L^2}{A^m}G_2^2
$$

Rearranging the inequality above, we obtain:

$$
\mathbb{E}\left[\left\|\nabla f^{s_m}(\boldsymbol{w}^{s^{(r)}_m})\right\|^2\right] \leq \frac{\mathbb{E}\left[f^{s_m}(\boldsymbol{w}^{s^{(r)}_m})\right]}{\eta} - \frac{\mathbb{E}\left[f^{s_m}(\boldsymbol{w}^{s^{(r+1)}_m})\right]}{\eta}
$$
$$
+ 2\eta L^3\left(B^2 + 1\right)\left(f^{s_m}(\boldsymbol{w}^{s^0_m}) - f^{s^\star_m}\right) + \left(1 - \frac{A^m}{K}\right)\frac{\eta L^2}{A^m}G_2^2
$$
$$
+ \sqrt{2G_1^2(G_2^2 + 2LB^2(f^{s_m}(\boldsymbol{w}^{s^{(r)}_m}) - f^{s^\star_m}))d^{c^{(r)}_m}}
$$

By defining $F^{s_m} := \max_r \{f^{s_m}\left(\boldsymbol{w}^{s^{(r)}_m}\right) - f^{s^\star_m}\}$, and then applying Lemma 2 with $q_1 = 2L^3\left(B^2 + 1\right)\left(f^{s_m}(\boldsymbol{w}^{s_m}) - f^{s^\star_m}\right) + \left(1 - \frac{A^m}{K}\right)\frac{L^2}{A^m}G_2^2$ to the first two terms while averaging the summation over the third term, we obtain.

$$
\mathbb{E}\left[\left\|\nabla f^{s_m}(\boldsymbol{w}^{s^{(R)}_m})\right\|^2\right] \leq \frac{f^{s_m}(\boldsymbol{w}^{s^0_m})}{\eta_{max}(R+1)}
$$
$$
+ \frac{2\sqrt{\left[2L^3\left(B^2 + 1\right)F^{s^0_m} + \left(1 - \frac{A^m}{K}\right)\frac{2L^2}{A^m}G_2^2\right]f^{s_m}(\boldsymbol{w}^{s^0_m})}}{\sqrt{R+1}}
$$
$$
+ \frac{1}{R+1}\sqrt{2G_1^2(G_2^2 + 2LB^2 F^{s^0_m})\sum_r d^{c^{(r)}_m}}
$$

According to Assumptions 6, the convergence of $\sum d^{c^{(r)}_m}$ implies the convergence of the third term. Consequently, the right term is bounded and converges as the number of rounds $R$ increases, thereby concluding the proof. By employing asymptotic notation, we derive the following expression for the server-side convergence rate:

$$
\mathbb{E}\left[\left\|\nabla f^{s_m}(\boldsymbol{w}^{s^{(R)}_m})\right\|^2\right] = \mathcal{O}\left(\frac{C_1}{R} + \frac{H_2\sqrt{F^{s^0_m}}}{\sqrt{RA^m}} + \frac{F^{s^0_m}}{\eta_{max}R}\right)
$$

where, $H_2^2 := L^3\left(B^2 + 1\right)F^{s^0_m} + \left(1 - \frac{A^m}{K}\right)L^2 G_2^2$ , $F^{s^0_m} := f^{s_m}\left(\boldsymbol{w}^{s^0_m}\right)$, and $C_1 = G_1\sqrt{G_2^2 + 2LB^2 F^{s^0_m}\sum_r d^{c^{(r)}_m}}$.

### B.3.2 NON-CONVEX FUNCTIONS

The convergence rates for client-side non-convex functions can be determined using techniques similar to those employed in the previous section. The corresponding convergence rate can be expressed as follows, using Lemma 2:

$$
\mathbb{E}\left[\left\|\nabla f^{c_m}(\overline{\boldsymbol{w}^{c_m}}^R)\right\|^2\right] = \mathcal{O}\left(\frac{H_1\sqrt{F^{c^0_m}}}{\sqrt{RA^m}} + \frac{F^{c^0_m}}{\eta_{max}R}\right)
$$

where, $F^{c_m^0} := f^{c_m} \left( \boldsymbol{w}^{c_m^0} \right)$.

The convergence rates for server-side non-convex functions can be determined using techniques similar to server-side convex functions. The resulting convergence rate is as follows:

$$\mathbb{E}\left[ \left\| \nabla f^{s_m}(\boldsymbol{w}^{s_m^{(R)}}) \right\|^2 \right] = \mathcal{O}\left( \frac{C_2}{R} + \frac{H_2\sqrt{F^{s_m^0}}}{\sqrt{RA^m}} + \frac{F^{s_m^0}}{\eta_{max}R} \right)$$

where, $C_2 = G_1\sqrt{G_2^2 + B^2G_1^2\sum_r d^{c_m^{(r)}}}$.

