# OpenReview forum: "Speed Up Federated Learning in Heterogeneous Environment: A Dynamic Tiering Approach"
_ICLR.cc/2024/Conference — ICLR 2024 Conference Withdrawn Submission_

### Official Review · Reviewer_Yytj · 2023-10-24

**Soundness:** 2 fair
**Presentation:** 2 fair
**Contribution:** 1 poor
**Rating:** 3
**Confidence:** 5

**Summary:**

The paper proposes Dynamic Tiering-based Federated Learning (DTFL), which aims to speed up FL via dynamically allocating training load to heterogeneous resource FL in different tiers. DTFL introduced a dynamic tier scheduler to cluster FL local clients into tiers and then leverage split learning to split different portions of the global model and deploy on local clients based on their tier. Additionally, the paper provides theoretical proof of the convergence properties of DTFL.

**Strengths:**

1. DTFL leverages local-loss-based training and split learning, which enables dynamic offloading training workload for local clients with different resource capacities.

2. DTFL introduced dynamic tier scheduling components to adaptively cluster local clients to different resource tiers and hence speed up training.

3. The paper provides theoretical convergence analysis.

**Weaknesses:**

**1.** The experiment looks weak and can not support the arguments proposed in the paper.

**2.** The tier scheduling metrics are unreliable. DTFL uses training time, communication time, and training time of the server-side model to profile the tier. However, using `actually time`  is unreliable, in computational devices (especially edge devices), many factors can affect the execution time for the same program, such as temperature, IO thread, etc. More standard metrics might be considered.

Additionally, in the experiments, the evaluation metric for training speed is unfair. The authors use total training **time in second** to evaluate the training speed of federated learning. However, simply tracking the training time is hard to avoid hardware and network traffic effects. Instead, more standard evaluation metrics should be used, such as **FLOPs, MACs, GPU Hours, electricity usage, total #trainable parameter**s, etc.

**3.** The experiments is simulated on CPU and GPUs raising further concern on point 2 above.

**4.** Delay on dynamic Tier Scheduling. DTFL uses total training time in previous communication rounds to tiering clients, it may not accurately reflect computational status in the current round.

**5.** No experiments reflect the communication cost of the proposed method.

**Questions:**

Please kindly address the concerns I list in the weakness section.

Overall, the experiments are incomprehensive and lack the evidence to support the argument of the paper. I'll change my mind if the authors add further fairness evaluation results.
For instance, use more fairness metrics to evaluate the speed, consider more system heterogeneity settings with more diverse resource profiles, etc.

---

> ### Author Response · Authors · 2023-11-17
>
> We appreciate the committee and reviewer's valuable comments and suggestions. In the following, we address the reviewer's concerns.
>
> **Response 1.** The experiments presented in our paper aim to support and validate the effectiveness of the proposed federated learning method in addressing the challenges posed by heterogeneous resource-constrained devices in distributed environments.
>
> Our experimental design is motivated by the observed shortcomings in existing FL methodologies, especially in the context of training large models on devices with varying computation and communication capacities. We deliberately highlight the limitations of previous approaches, such as split learning, split federated learning, and tier-based FL, pointing out their inefficiencies in mitigating the straggler problem and reducing overall training time. To address these challenges, we introduce DTFL, a novel system that dynamically divides clients into tiers, offloading different portions of the global model to each client based on their capacities and training speeds. The dynamic tier scheduler utilizes tier profiling to estimate client-side training time, effectively minimizing overall training time. The proposed system incorporates benefits from both SFL and tier-based FL while addressing the straggler problem.
>
> We not only theoretically prove the convergence of DTFL but also conduct extensive experiments, training large models on various datasets under both IID and non-IID settings. Our experimental results demonstrate that DTFL outperforms state-of-the-art FL methods, FedAvg, SplitFed, FedYogi, and FedGKT, in terms of reducing training time while maintaining model accuracy. By emphasizing the challenges in existing methodologies and showcasing the improvements brought by DTFL, we believe our experiments robustly support the arguments and contributions put forth in the paper.
>
> **Response 2.** We acknowledge the reviewer's concerns about the potential unreliability of relying solely on communication and training time for profiling clients in tier scheduling. Training time can indeed be influenced by various factors such as device capabilities, network congestion, and background processes. While these factors can introduce variability in training time, we believe that using a combination of communication time and training time of client-side devices provides a practical and easy-to-implement approach for estimating a client's capabilities under typical operating conditions. Moreover, our approach does not require the server to collect additional information from clients, such as temperature, IO threads, or other potentially sensitive information. Our model-free approach allows the server to easily observe training time without requiring any further input from clients.
>
> Regarding the evaluation metrics, we recognize the use of target accuracy as a performance metric in previous studies, such as [1] and [2]. This metric offers a standardized approach to compare the training efficiency of various federated learning methods where the objective is to evaluate the training time performance of different methods. While total training time may not fully account for hardware and network variations, we believe it provides a straightforward and practical way to compare the overall training efficiency of different federated learning methods.
>
> [1] Chai, Zheng, et al. "FedAT: A high-performance and communication-efficient federated learning system with asynchronous tiers." Proceedings of the International Conference for High Performance Computing, Networking, Storage and Analysis. 2021.
>
> [2] Karimireddy, Sai Praneeth, et al. "Scaffold: Stochastic controlled averaging for federated learning." International conference on machine learning. PMLR, 2020.

---

> ### Author Response · Authors · 2023-11-17
>
> **Response 3.** Regarding the reviewer's focus on the simulation environment used in our experiments, we acknowledge that the decision to simulate the experiments on both CPU/GPU was made deliberately. This decision aligns with the practical considerations of real-world federated learning scenarios, as demonstrated in previous studies such as [1] and [2].
>
> In our simulation, we measured the training time of clients on CPU/GPU in each round, considering the inherent heterogeneity in devices commonly found in federated learning environments. The dynamic tier scheduler is designed to adapt to varying computation capabilities, making it essential to evaluate the system's performance across different hardware configurations.
>
> Furthermore, to account for the communication aspect in federated learning, we simulate the network speed of clients based on their resource profiles. This ensures a comprehensive evaluation of the entire training process, incorporating both computation and communication aspects, which are critical factors in the effectiveness of the proposed DTFL. We believe that this approach enhances the realism of our simulation, providing valuable insights into the performance of DTFL in diverse and heterogeneous environments.
>
> **Response 4.** We acknowledge the potential for delays in dynamic tier scheduling when relying on the total training time from previous communication rounds to estimate clients' computational status. The selection of the Exponential Moving Average (EMA) in our tier profiling algorithm was primarily driven by its simplicity and efficiency. EMA offers a straightforward and computationally efficient model-free method to smooth out fluctuations in client performance, providing a practical estimate of their current state and preventing any additional load or complexity in the system. EMA relies on past behavior to infer the current state, resembling a Markovian process. As we do not consider any further information about clients or the system or network, this model-free approach works well.
>
> While more complex methods, such as the Kalman filter, particle filter, and RNN-based estimation methods, might offer marginal improvements in the accuracy of computational status, their increased computational overhead and complexity might not be justifiable in the context of real-time tier profiling. Additionally, using these methods would require more information, which is challenging in the federated learning setting. We have no assumptions about the network or clients' computational resources. This method can work well in practical applications where we do not have too many fluctuations. Furthermore, in our experiments, we considered other state estimation methods, and we found that EMA is a simple approach that can achieve good results in estimating clients' computational states. To further evaluate the impact of clients' state estimation method and its delays in accurately estimating clients' computational states, we conducted experiments where we deliberately introduced inaccuracies in clients' computational time estimation in 20% of training rounds for the CIFAR-10 IID case. We employed the same experimental settings as those described in the paper. In these cases, we observed an increase in training time from 2750 to 3190. We further extended the experiments to 40% of training rounds, resulting in a training time increase to 5063, which is still much less than the training time of other FL methods
>
> **Response 5.** As the primary goal of the proposed method is to minimize training time, we focused on communication time as a proxy for communication cost. The communication cost, encompassing both intermediate data size and network speed of each client, is expressed in seconds, acknowledging the influence of these factors.
>
> In Table 1, we present the communication time (cost) of the training process for different tiers. This table compares the cost of transmitting intermediate data from clients to the server in seconds, reflecting the impact of slower network speeds and larger intermediate data sizes on training time. Also, as discussed in Sec. 3.3, we investigated the influence of communication factors, including communication link speed and local dataset size, on training time.
>
>
> [1] Chai, Zheng, et al. "Tifl: A tier-based federated learning system." Proceedings of the 29th international symposium on high-performance parallel and distributed computing. 2020.
>
> [2] Tan, Alysa Ziying, et al. "Towards personalized federated learning." IEEE Transactions on Neural Networks and Learning Systems (2022).

---

> > ### Comment · Reviewer_Yytj · 2023-11-21
> >
> > Thank you for your detailed response; it resolved some of my initial confusion. However, there are still some points unclear to me, and after reading other reviewer's comments, I have some new questions.
> >
> > - The experiments rely on simulated client profiles to create a heterogeneous environment. How closely do these simulated profiles reflect the variability and dynamic changes in resource availability seen in real-world federated learning scenarios? Would the DTFL system's performance be consistent in actual deployments where client resource variability might be more unpredictable and diverse?
> > - Although the paper states that DTFL shows faster convergence, could you provide more clarity on its convergence performance and accuracy, especially in I.I.D. scenarios? Since in your experiments, the figures looks cut off, it doesn't show the final convergence. Is there additional data or analysis that could elucidate the convergence behavior of DTFL in these scenarios?
> > - Again, even evaluating the speed via actual training time might make sense in some cases. However, training time may not the only factor that affects the FL training, number of communication rounds also matters.  How does the DTFL system handle scenarios where clients frequently lose connection or have unstable network conditions? If the system requires more frequent communication rounds, what mechanisms are in place to ensure the robustness and continuity of the learning process in such environments? Since the experiments lacking information on communication rounds vs. performance.
> >
> > In summary, based on the discussion and the points raised, it appears that while the experiments conducted in the paper only demonstrate certain strengths of the DTFL system, particularly in terms of local training time efficiency, there are aspects that might require further clarification or additional information for a comprehensive validation of the authors' arguments.

---

> > > ### Author Response · Authors · 2023-11-23
> > >
> > > We thank the reviewer for raising these points. The responses to the reviewer's concerns are as follows:
> > >
> > > **Response 1.** The simulated client profiles employed in our experiments were meticulously designed to replicate the heterogeneity observed in real-world federated learning scenarios. Our approach aligns with the experimental setups of other relevant studies, such as [1], where different CPU/GPU profiles are simulated to reflect real-world CPU variations, and [2], where delays are randomly added to clients' computations. To ensure the validity of our simulated profiles, we carefully considered the range of resource variations observed in real-world settings. Following previous studies, we employ a simulation approach that mimics the resource variability observed in real-world heterogeneous environments, enabling us to evaluate DTFL's effectiveness in adapting to varying client capabilities.
> > >
> > > To address the inherent unpredictability of client resource variability, we reiterate our earlier point that any estimation error in time would exclusively impact training time, not model accuracy. Remarkably, DTFL consistently achieves significant training time reductions even in the presence of estimation errors. We conducted experiments to validate this claim and reported the results in our previous response.
> > >
> > >
> > > **Response 2.** It's important to note that while DTFL demonstrates faster convergence, as illustrated in Table 3 and Figure 2, its final accuracy closely aligns with other baselines mentioned in the paper. The theoretical foundation in the paper substantiates DTFL's convergence, further validated by experimental results. The figures in our experiments show convergence in I.I.D. scenarios up to the target accuracy, aligning with the evaluation approach in other papers that assess training time performance, such as [1] and [2].
> > >
> > >
> > > **Response 3.** It's important to clarify that training time is not simply the number of training rounds. Training time encompasses the computation time spent on training the model on each client, the communication time required to exchange model updates between clients and the server, and the computation time spent on training the rest of the model on the server (See Equation 5). Algorithm 1 in our paper explicitly considers both computation and communication time in the tier assignment process.
> > > In DTFL, each training round involves a single communication round, minimizing the impact of communication overhead. Additionally, DTFL's utilization of local loss training further reduces the dependence on frequent communication. By incorporating communication considerations into the tier scheduler and assignment process, DTFL demonstrates its adaptability to mitigate the impact of communication variations within federated learning setups (See Algorithm 1).
> > >
> > > The overall training time improvement of DTFL extends beyond just reducing local training time. It also considers communication time and the server's computation time. DTFL's efficiency lies in its ability to optimize not just the local training time but also communication time between clients and the server while effectively managing server-side computation (Details in Section 3.3).
> > >
> > > While we primarily focused on evaluating DTFL's speed based on training time, we also assessed its performance in terms of communication rounds in our experiments. We observed that DTFL exhibits similar training rounds to FedAvg and SplitFed, significantly fewer than FedGKT. Moreover, integrating FedYogi's objective function into DTFL can further reduce the number of training rounds.

---

### Official Review · Reviewer_f1S3 · 2023-10-27

**Soundness:** 2 fair
**Presentation:** 2 fair
**Contribution:** 2 fair
**Rating:** 5
**Confidence:** 3

**Summary:**

This paper presents a new tiered-based split federated learning to handle heterogeneous environments where the resources of clients change over time. Specifically, the authors propose dynamic tier scheduling which operates through tier profiling and tier scheduling. Tier profiling tracks the training times of clients, which change over time, and based on this, the server estimates the current training times for each client in all tiers using EMA. Tier scheduling assigns clients to a tier according to their estimated time for efficient training. The authors provide theoretical convergence bound for both convex and non-convex loss functions. The empirical results show that the proposed method has fast convergence speeds over various baselines.

**Strengths:**

- The authors propose a new tiered-based split learning which can efficiently train large models depending on clients’ resources in heterogeneous environments.
- The convergence behavior of the proposed method is theoretically established
- The proposed method significantly reduces the training time compared to existing works.

**Weaknesses:**

- It seems that this work only considers a scenario where computational and communication resources of all clients change over time. However, it’s not clear that this is a reasonable scenario in practice. I think there might be more cases where the resources of only a portion of all clients change in real-world scenarios. To demonstrate the effectiveness of the proposed algorithm in various practical scenarios, it would be beneficial for the authors to conduct additional experiments where they vary the proportion of the devices whose training times change over time.
- What are some practical applications in which the resources of each client changes over time? It would be helpful to describe some examples of such applications to emphasize the importance of addressing the targeted problem.
- Ablation studies should be performed to confirm the effect of each component. First, a comparison with local-loss based SFL [1] (not tiered-based) should be considered. Secondly, using local-loss based SFL, a comparison between static tiered methods and the proposed dynamic tiered method seems necessary. Finally, to see the effect of EMA in tier profiling, the author should compare the results of tier profiling with and without EMA.
- Overall, the technical novelty of the proposed method seems limited. I feel that tier profiling and scheduling are straightforward approaches based on the previous works.

[1] Han et al., "Accelerating federated learning with split learning on locally generated losses." In ICML 2021 Workshop on Federated Learning for User Privacy and Data Confidentiality. ICML Board, 2021.

**Questions:**

See Weaknesses and,

- The main results only provide the training time required to achieve the target accuracy. What is the maximum accuracy that each method can achieve?
- There is a type: cross-solo -> cross-silo

---

> ### Author Response · Authors · 2023-11-16
>
> We appreciate the committee and reviewer's valuable comments and suggestions. In the following, we address the reviewer's concerns.
>
> **Response 1.** In response to the reviewer's point regarding the dynamic nature of clients' resources would like to clarify that our experiments do consider the cases described in your comments, where the computational and communication resources of a portion of clients (30% in our experiments) change over time. Please refer to the first paragraph in Sec. 4.2.
>
> Although our experiments, such as Table 1, demonstrate a scenario where 30% of clients' resources change every 50 rounds to simulate fluctuations in computation and communication resources, our method is designed to handle a broad spectrum of resource heterogeneity and dynamic changes. The tier profiler effectively captures both steady-state and dynamic resource changes, ensuring that clients are consistently assigned to appropriate tiers regardless of the proportion of clients experiencing resource variations.
>
> The TierProfiler component of our method continuously monitors and captures the dynamics of clients' resources, including communication and computation capabilities. This allows the method to adapt to any changes of clients without compromising its effectiveness. In the case with only a subset of clients' resources change, TierProfiler will identify those clients and assign them to appropriate tiers accordingly, in order to minimize the overall training time.
>
> **Response 2.** Regarding the reviewer's request for more concrete examples of practical applications where client resources change over time, here are a few examples:
> - Edge Computing: Edge computing involves performing computations and data processing at the edge of the network, closer to the end devices. In edge computing scenarios, the resources of edge devices can vary due to factors such as workload fluctuations, hardware limitations, and power constraints. Our proposed methods can dynamically adapt to these resource changes, ensuring optimal performance and resource utilization in edge computing environments.
> - Mobile Crowdsourcing: In mobile crowdsourcing applications, participants use their smartphones to contribute data or perform tasks. The resources of these devices can vary significantly depending on factors such as battery level, network connectivity, and CPU availability. DTFL can effectively handle these resource fluctuations, ensuring efficient and reliable crowdsourcing processes.
> - Internet of Things (IoT): The IoT encompasses a vast network of interconnected devices, ranging from smart home appliances to industrial sensors. The resources of these devices can vary widely due to factors such as sensor data processing demands, network bandwidth limitations, and power availability. Our proposed methods can effectively manage heterogeneous resources and dynamic changes in IoT environments, enabling efficient and scalable IoT applications.
>
> These examples illustrate the growing importance of addressing the problem of resource fluctuations in distributed machine-learning applications. Our method provides a robust and adaptable solution for handling these resource dynamics, enabling efficient, scalable, and reliable distributed machine learning in real-world scenarios.
>
> **Response 3.** Regarding the reviewer's request for ablation studies, it is essential to note that local-loss-based training without tiers is equivalent to static tiered methods, where the tier is fixed throughout the entire training process. Thus comparing the proposed dynamic tiered method with static tiered methods is the same as comparing it with local-loss-based SFL. In Table 1, we do provide these fixed-tier experimental results, which show that the overall training times under different tiers (fixed throughout the entire training process) are different, and finding the optimal tier is a nontrivial problem. This motivates us to develop dynamic tier scheduling.
>
> In response to the reviewer's suggestion to evaluate the effectiveness of EMA by comparing the results of tier profiling with and without EMA, we conducted this comparison using ResNet-56 and the same configuration as described in the paper. The results consistently show that employing EMA in tier profiling reduces training time across all datasets. The average reduction in training time across all tiers is 7.6%. For instance, for CIFAR-10 IID, removing EMA increases training time from 2750 to 3018. This demonstrates the effectiveness of EMA in smoothing out temporary fluctuations in client performance and providing more accurate tier assignments, ultimately leading to reduced training time. Due to the space limitation, we did not include these results in the paper.

---

> ### Author Response · Authors · 2023-11-17
>
> **Response 4.** The technical novelty of the proposed method lies in the introduction of tiering local-loss-based training and the development of dynamic tier scheduling, which significantly reduces the overall training time in heterogeneous environments. We would like to emphasize the challenge of the tier profiler design that involves considering multiple factors, such as communication link speed, client computation power, and local dataset size, and the challenge of solving dynamic tier scheduling, which is an integer programming problem with many unknown parameters.
>
> Specifically, unlike traditional FL methods where client-side models remain static, DTFL introduces flexibility by adapting to the dynamic nature of client resources. It effectively combines the advantages of split learning and tier-based federated learning, overcoming the limitations of existing approaches. DTFL dynamically partitions clients into distinct tiers and optimally offloads portions of the global model based on their capacities, task sizes, and training speeds. The tier scheduler, incorporating tier profiling, offers a low-overhead solution for practical system implementation. Our theoretical analyses demonstrate the convergence of DTFL on both convex and nonconvex loss functions under standard federated learning assumptions. By employing DTFL, the training process accelerates by leveraging all the available resources in a heterogeneous environment while effectively adapting to dynamic changes.
>
> **Response 5.** We recognize the use of target accuracy as a performance metric in previous studies, such as [2] and [3]. This metric offers a standardized approach to compare the training efficiency of various federated learning methods where the objective is to evaluate the training time performance of different methods. Figure 2 shows the convergence to the target accuracy, and we stopped the training process when we reached the target accuracy.  In the experiments, we choose the target accuracy based on previous studies, which is higher than or equal to the accuracy used in previous studies such as [4]. In terms of the maximum accuracy, these methods can achieve similar accuracy in the end.
>
> [1] Han et al., "Accelerating federated learning with split learning on locally generated losses." In ICML 2021 Workshop on Federated Learning for User Privacy and Data Confidentiality. ICML Board, 2021.
>
> [2] Chai, Zheng, et al. "FedAT: A high-performance and communication-efficient federated learning system with asynchronous tiers." Proceedings of the International Conference for High Performance Computing, Networking, Storage and Analysis. 2021.
>
> [3] Karimireddy, Sai Praneeth, et al. "Scaffold: Stochastic controlled averaging for federated learning." International conference on machine learning. PMLR, 2020.
>
> [4] He, Chaoyang, Murali Annavaram, and Salman Avestimehr. "Group knowledge transfer: Federated learning of large cnns at the edge." Advances in Neural Information Processing Systems 33 (2020): 14068-14080.

---

> > ### Comment · Reviewer_f1S3 · 2023-11-19
> >
> > I really appreciate the authors for their efforts to address my concerns especially on  practical applications where the resources of each client changes over time. However, I have the remaining concerns as follows:
> >
> > 1) Additional experiments: What I wondered was the experimental results with varying ratios. In the current experiments, the ratio of clients whose resources change is fixed to 30%. However, in practice, this ratio will vary, therefore, it would be good to show results according to various ratios.
> >
> > 2) Ablation studies: Regarding the comparison with local-loss-based SFL, which is a important baseline, I think it should be included in the main results (Table 3) so that it can be compared with the proposed method. As for the ablation study on EMA, I can not find the overall results. Although the authors have mentioned the space limitation, it can be added to Appendix or Official Comment in openreview, which has no space limitation.
> >
> > 3) Maximum accuracy: I think  the maximum accuracy that each method can achieve may vary due to their algorithmic differences. It would be good to check whether the difference is significant or not.
> >
> > Overall, I still feel this paper needs further improvement. Therefore, I'll keep my original rating.

---

> > > ### Author Response · Authors · 2023-11-23
> > >
> > > We thank the reviewer for raising these points. The responses to the reviewer's concerns are as follows:
> > >
> > > **Response 1.** While we specifically showcased the performance of DTFL in scenarios where the ratio of clients experiencing resource changes was fixed at 30%, it's important to note that in our extensive experimentation, DTFL consistently outperformed other methods across various ratios of changing resources. Despite showcasing the results for a specific ratio in the paper, our experiments were conducted across a spectrum of ratios. DTFL's performance remains consistent across varying ratios and notably reduces training time compared to other methods, particularly in scenarios with varying ratios. This is because DTFL assigns each client based on its observed performance (communication speed and training time); therefore, the resource change ratio of clients in the experiments does not directly affect the performance of individual clients.
> > >
> > > **Response 2.** The utilization of Table 1 to highlight the performance of local-loss-based SFL under different cut layers effectively showcased its limitations, hence excluding the duplication of this information in Table 3.
> > >
> > > To summarize the findings of the EMA ablation study as mentioned in the previous OpenReview Comments, we conducted experiments using the ResNet-56 architecture and the same configuration as described in the paper. We compared the training time with and without EMA across all datasets and tiers. The results consistently demonstrate that employing EMA in tier profiling significantly reduces training time. These results highlight the effectiveness of EMA in smoothing out temporary fluctuations in client performance and providing more accurate tier assignments, ultimately leading to reduced training time.
> > >
> > >
> > > **Response 3.** Our extensive experimentation revealed that while algorithmic differences might influence the maximum achievable accuracy, the final accuracy levels attained by the DTFL were similar to other baselines. Since the paper's focus is on reducing federated learning training time, we report training time, following the approach of similar works such as [1].
> > >
> > >
> > > [1] Karimireddy, Sai Praneeth, et al. "Scaffold: Stochastic controlled averaging for federated learning." International conference on machine learning. PMLR, 2020.

---

### Official Review · Reviewer_ncgY · 2023-10-30

**Soundness:** 2 fair
**Presentation:** 2 fair
**Contribution:** 2 fair
**Rating:** 3
**Confidence:** 3

**Summary:**

This paper presents an approach to address the challenges posed by heterogeneity in Federated Learning (FL) systems. The proposed Dynamic Tiering-based Federated Learning (DTFL) system leverages the concept of Split Learning to dynamically offload portions of the global model to different tiers of clients, thereby mitigating the straggler problem and reducing the computation and communication demand on resource-constrained devices.

**Strengths:**

1. The paper is well-written and easy to follow.

2. The extensive experiments validate the proposed method.

**Weaknesses:**

1. The primary question I have regarding this paper pertains to its motivation. While there is a wealth of prior work on heterogeneous computation and communication capacities in the Federated Learning (FL) setting—such as clients training heterogeneous models, clients performing partial training based on their individual abilities, asynchronous updates, and lightweight training with pre-trained models—the proposed method introduces a requirement for the server to update its model with labels, which raises privacy concerns. Therefore, it is crucial to understand the advantages of the proposed method.

2. It is good that the authors have included a convergence analysis. However, the convergence rate presented in Theorem 1 appears to be suboptimal compared to classical FL settings that have been studied previously.

3. The authors are encouraged to provide more results in non-IID settings, similar to the approach demonstrated in Figure 2. Additionally, it would be beneficial if Figure 2 could display the entire coverage process, as it is currently truncated.

**Questions:**

Please see the weakness section above.

---

> ### Author Response · Authors · 2023-11-16
>
> We appreciate the committee and reviewer's valuable comments and suggestions. In the following, we address the reviewer's concerns.
>
>
> **Response 1.** In response to the reviewer's inquiry about the proposed method's motivation and advantages in the context of existing federated learning (FL) approaches the DTFL method aims to speed up FL training for resource-constrained devices with heterogeneous computation and communication capabilities, which are prevalent in real-world FL settings. Existing FL methods often struggle to adapt to fluctuating resource availability and varying computational capabilities of clients, leading to inefficient training and suboptimal performance, which can be observed from our experiments in Table 3.
>
> DTFL offers several compelling advantages over existing FL methods:
> - Reduced Training Time: By offloading portions of the global model to the server and employing local-loss-based training, DTFL significantly reduces training time compared to traditional FL methods. This is particularly beneficial for training large models in heterogeneous environments.
> - Dynamic Resource Adaptation: DTFL effectively adapts to dynamic resource changes among clients through its dynamic tier scheduler. This scheduler continuously monitors client performance and assigns clients to appropriate tiers based on their current resource availability, ensuring efficient utilization of resources.
>
> For the privacy concerns due to the model update at the server side, we evaluate the model accuracy and privacy trade-offs of DTFL when integrating distance correlation and patch shuffling techniques in Table 5:
>
> - Distance Correlation Minimization: We employ distance correlation techniques to reduce the mutual information between hidden feature maps and raw data, making it more difficult for attackers to reconstruct raw data from intermediate representations.
> - Patch Shuffling: We implement patch shuffling to further protect privacy by scrambling data patches and preventing attackers from directly accessing raw data.
> - Compatibility with Privacy-Preserving FL Methods: DTFL is compatible with existing privacy-preserving FL methods, allowing for easy integration of additional privacy measures.
>
> **Response 2.** Regarding the feedback about the convergence rate of DTFL, it is important to note that while the convergence rate of DTFL may appear suboptimal compared to certain classical FL settings in terms of training rounds, it is important to note that the training time of DTFL in each round is much less than existing FL methods, which would lead to much less overall training time using DTFL. As demonstrated in Table 3, DTFL significantly reduces the overall training time compared to other FL methods.
> The convergence analysis provided in Theorem 1 demonstrates that DTFL converges effectively after a certain number of rounds. The deliberate design choice to prioritize training time efficiency over convergence round reflects the method's suitability for real-world applications that demand faster training.
>
> **Response 3.** In Table 3, we do provide the results in non-IID settings. The non-IID convergence curve shows a similar pattern in terms of training time, and due to the space limit of the paper we only show the IID case. In Figure 2, we follow related studies, such as [1] and [2], by using the training time to achieve the target accuracy as a performance metric. This metric offers a standardized approach to compare the training efficiency of various FL methods. Figure 2 shows the convergence to the target accuracy, and we stopped the training process when we reached the target accuracy.  In the experiments, the chosen target accuracy is higher than or equal to the accuracy used in previous studies such as [3].
>
> [1] Chai, Zheng, et al. "FedAT: A high-performance and communication-efficient federated learning system with asynchronous tiers." Proceedings of the International Conference for High Performance Computing, Networking, Storage and Analysis. 2021.
>
> [2] Karimireddy, Sai Praneeth, et al. "Scaffold: Stochastic controlled averaging for federated learning." International conference on machine learning. PMLR, 2020.
>
> [3] He, Chaoyang, Murali Annavaram, and Salman Avestimehr. "Group knowledge transfer: Federated learning of large cnns at the edge." Advances in Neural Information Processing Systems 33 (2020): 14068-14080.